# Ice supersaturated regions: properties and validation of ERA-Interim Reanalysis with IAGOS in-situ water vapour measurements

Philipp Reutter[1], Patrick Neis[1,2,*], Susanne Rohs[2], and Bastien Sauvage[3]

[1]Institute for Atmospheric Physics, Johannes Gutenberg University Mainz, Mainz, Germany
[2]Institute of Energy and Climate Research Troposphere (IEK-8), Forschungszentrum Jülich, Jülich, Germany
[3]Laboratoire d'Aerologie, Université de Toulouse, CNRS, UPS, Toulouse, France
[*]now at: CGI Deutschland B. V. CO. KG, Frankfurt, Germany

**Correspondence:** Philipp Reutter (preutter@uni-mainz.de)

**Abstract.** Cirrus clouds and their potential formation regions, so-called ice-supersaturated regions (ISSRs) with values of relative humidity with respect to ice exceeding $100\%$ occur frequently in the tropopause region. It is assumed that ISSRs and cirrus clouds can change the tropopause structure by diabatic processes, driven by latent heating due to phase transition and interaction with radiation. For many research questions a three-dimensional picture including a sufficient temporal resolution of the water vapour fields in the tropopause region is required. This requirement is fulfilled nowadays by reanalysis products such as the European Centre for Medium-Range Weather Forecasts (ECMWF) ERA-Interim reanalysis. However, for a meaningful investigation of water vapour in the tropopause region a comparison of the reanalysis data with measurement is advisable, since it is difficult to measure water vapour and to assimilate meaningful measurements into reanalysis products. Here, we present an intercomparison of high-resolution in-situ measurements aboard passenger aircraft within the European Research Infrastructure IAGOS (In-service Aircraft for a Global Observing System; http://www.iagos.org) with ERA-Interim. Temperature and humidity data over the North Atlantic from 2000 to 2009 are compared relative to the dynamical tropopause. The comparison of the temperature shows a good agreement between measurement and ERA-Interim. While ERA-Interim also shows the main features of the water vapour measurements of IAGOS, the variability of the data is clearly smaller in the reanalysis data set. The combination of temperature and water vapour leads to the relative humidity with respect to ice ($\mathrm{RH_i}$). Here ERA-Interim deviates from the measurements concerning values of larger than $\mathrm{RH_i} = 100\%$, both in number and strength of supersaturation. Also pathlengths of ISSRs along flight tracks are investigated, representing macrophysical properties as linked to atmospheric flows. The comparison of ISSR pathlengths shows distinct differences, which can be traced back to the spatial resolution of both data sets. Also, the seasonal cycle and height dependence of pathlengths changes for the different data sets due to their spatial resolution. IAGOS shows significantly more smaller ISSRs compared to ERA-Interim. A good agreement begins only at pathlengths in the order of the ERA-Interim spatial resolution and larger.

# 1 Introduction

Water vapour is the most important greenhouse gas in the atmosphere and therefore plays a major role in the Earth's radiative balance (Myhre et al., 2013). Especially in condensed form water is also of large significance for the planetary radiation. Clouds can reflect incoming solar radiation, while absorbing and reemitting longwave radiation from the earth. Particularly the
effect of cirrus clouds is still challenging. Whether a cirrus cloud has a net warming or cooling effect on the Earth's atmosphere depends strongly on altitude, available humidity and microphyiscal properties like number, size and type of ice nuclei (IN). Even the same exact cirrus cloud can change the sign of its net forcing depending on the time of day (Joos et al., 2014). Beside natural cirrus clouds also the aircraft-induced contrail cirrus clouds play an important role for the radiative budget (Kärcher, 2018).
The control parameter for cold cloud formation in the upper troposphere is relative humidity with respect to ice, which reaches supersaturation by exceeding the temperature dependent water-holding capacity of the air mass (Gierens and Spichtinger, 2000; Spichtinger et al., 2003b).

$$\mathrm{RH_i} = 100 \cdot \frac{p_\mathrm{v}}{p_\mathrm{si}(T)} \tag{1}$$

where $p_\mathrm{v}$ is the present water vapour partial pressure and $p_\mathrm{si}$ the water vapour saturation pressure over ice water at temperature
$T$, respectively. The amount of ice supersaturation needed to form ice crystals depends strongly on the nucleation mechanism. Homogeneous nucleation of solution droplets requires supersaturations with respect to ice of at least $45\,\%$ (Koop et al., 2000), whereas heterogeneous freezing occurs at much lower supersaturations (DeMott et al., 2003; Mohler et al., 2006).

Ice supersaturation, first hypothesized by Alfred Wegener in 1911, is commonly found in the upper troposphere (Gierens et al., 1999; Spichtinger and Leschner, 2016; Gettelman et al., 2006). These so-called ice supersaturated regions (ISSR), i. e.
air masses in the status of ice supersaturation, constitute an important formation region for in-situ cirrus clouds (Krämer et al., 2016). While ISSRs alone do only have a minor effect on the local radiative budget (Fusina et al., 2007), the transformation from an ISSR to a region with cirrus clouds has a significant effect. Although ISSRs and cirrus clouds are mostly found in the upper troposphere, they also occur above the tropopause and have an effect on the lower stratosphere (LS). Since the region of the upper troposphere and lower stratosphere, the so-called UTLS region, is characterized by the coldest and driest air (Dessler
and Sherwood, 2009; Held and Soden, 2000) the outgoing long-wave radiation is most sensitive to absolute changes in the UTLS water vapour (Riese et al., 2012).

Besides the major role in the planetary radiation balance, water vapour distributions in the upper troposphere and lower stratosphere influences the UTLS chemistry. For example, stratospheric water vapour is partly a product of photochemical methane oxidation and will increase with anthropogenically increasing tropospheric methane concentrations (Rohs et al., 2006).
This increase of water vapour could lead to a more frequent formation of polar stratospheric clouds causing more ozone destruction in the stratosphere (Solomon et al., 2010). The chemical impact of tropospheric water vapour is, for example, the reaction with photolyzed ozone to the hydroxyl radical OH which further reacts with hundreds of gases and also leads to the rapid formation of acids that are deposit in precipitation (Thompson, 1992).

Another important aspect of water vapour distribution in the atmosphere is its feedback on atmospheric motions and stability. Water vapour is transported quickly through the atmosphere and redistributes energy by phase changes. For example, the condensation of water vapour close to the tropopause in potentially unstable layers can trigger the so-called shallow cirrus convection by latent heat release (Spichtinger, 2014). This alters the temperature and stability close to the tropopause with further implications on the exchange of air masses between troposphere and stratosphere.

Hence, a thorough description of processes related to the water vapour distribution is of crucial importance. However, measurements of water vapour at the tropopause level are not trivial. Beside radiosonde data the most important in-situ data set is provided by in-service passenger airplanes. Since 1994, commercial passenger aircrafts are measuring water vapour in the UTLS within the framework IAGOS (In-service Aircraft for a Global Oberserving System, (Petzold et al., 2015)) and its predecessors MOZAIC (Measurement of Ozone and Water Vapour on Airbus in-service Aircraft, (Marenco et al., 1998)) and CARIBIC (Civil Aircraft for the Regular Investigation of the Atmosphere Based on an Instrument Container, (Brenninkmeijer et al., 2007)). These regular measurements on a global scale are unique in their quantity, continuity and quality of measurements. Using five years of the continuous measurements over the North Atlantic, Gierens et al. (1999) described the humidity distribution in this region. These results were then used to improve the cloud scheme in the European Centre for Medium-Range Weather Forecast (ECMWF) Integrated Forecast Model (IFS) including the parameterization of superaturation with respect to ice in the cloud-free part of the grid box (Tompkins et al., 2007)

Meanwhile, the IAGOS data set now spans about 20 years and also allows trend analysis, for example with regard to the temperature. Here, a difference between in-situ and modelled data arise. While IAGOS exhibits a neutral temperature trend in the LS, the reanalysis data ERA-Interim of the ECMWF shows a temperature trend of $+0.56\,\mathrm{K\,decade^{-1}}$ (Berkes et al., 2017). This underlines the importance of a thorough comparison between measurements and modelled data. To investigate the mutual influence of water vapour in the UTLS region in the future, properties regarding the water vapour such like relative humidity with respect to ice, fraction of ice supersaturated regions or pathlengths of ISSRs are compared between IAGOS measurements and ERA-Interim output.

Although the new version of reanalysis data from the ECMWF, ERA5, is already available, we conducted this study using ERA-Interim. Many studies in this decade regarding the UTLS region are based on ERA-Interim model output (e.g. Zhan and Wang (2012); Riese et al. (2012); Uma et al. (2014); Madonna et al. (2014); Reutter et al. (2015)). Also, ERA-Interim is still used in many ongoing investigations. Therefore, a comparison between measurements and ERA-Interim is still valuable.

This study is part of a joint investigation of water vapour in the upper troposphere to the lower stratosphere. A companion study by Petzold et al. (2019) focuses on the physical interpretation of the water vapour distribution in the UTLS region. There, a detailed investigation of the seasonal cycle of $\mathrm{RH_i}$ and ISSRs, the physio-chemical signature of ISSR, the ISSR fraction and cirrus cloud occurrence is presented. Additionally, they also present a trend analysis.

The aim of our work is twofold. On the one hand, to assess the quality of the description of water vapour in the UTLS region in the ERA-Interim data set. To have a sufficient size of data, the data set includes 10 years from 2000 to 2009. On the other hand, we extend the physical investigation of Petzold et al. (2019) to characterize the horizontal scales of ISSRs as linked to

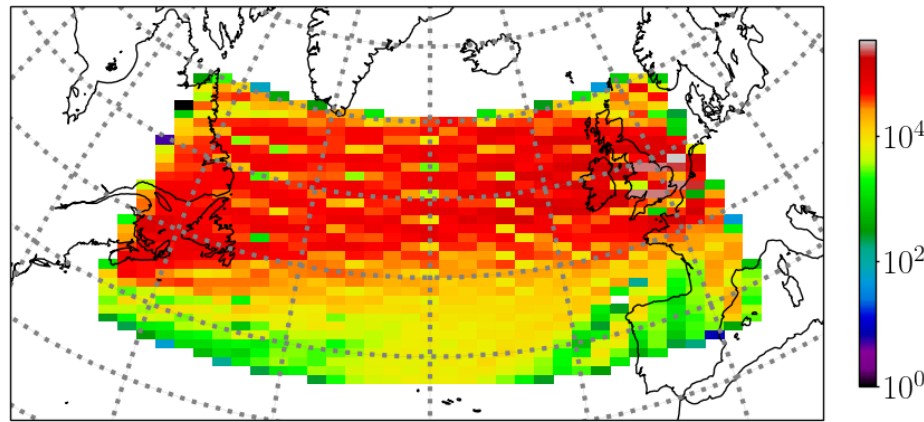

**Figure 1.** Number of IAGOS measurements per gridbox ($2°$ in longitudinal and $0.57°$ in latitudinal direction) during January 1st, 2000 to December 31st, 2009.

atmospheric flows and also depending on the seasonal cycle and height. We also compare these results to ERA-Interim as well as to other studies.

Sect. 2 describes the data sets and the methodology. In Sect. 3 the comparison of IAGOS and ERA-Interim is presented for the variables temperature, water vapour mixing ratio and relative humditiy with respect to ice. Sect. 4 investigates the horizontal
scales of ISSRs. The conclusion can be found in Sect. 5.

## 2   Data and methodology

The evaluation of the reanalysis data is based on in-situ measurements. Both data sets are presented in the following.

### 2.1   IAGOS

The European Research Infrastructure 'In-service Aircraft for a Global Observing System' (IAGOS, Petzold et al. (2015))
provides long-term in-situ measurements in the UTLS region. The IAGOS-CORE component, successor of the MOZAIC part, comprises the implementation and operation of autonomous instruments installed on long-range aircraft of internationally operating airlines for continuous, global-scale and daily measurements of reactive gases and long-lived greenhouse gases (e.g. $CO$, $CO_2$, $CH_4$ and water vapour), important chemically active trace gases (e.g. $O_3$), as well as aerosol, dust and cloud particles (Bundke et al., 2015; Filges et al., 2015).
Especially in the UTLS region these measurements are very valuable as most flight tracks are situated in heights between 9 to $13\,km$, depending on the actual weather conditions, seasons and geographic region.

Starting from August 1994 more than 60000 flights (May 2019) have been performed, including data from the IAGOS predecessor MOZAIC (August 1994 to December 2014) and data from the IAGOS project starting in July 2011 until present.

From Figure 1 it is obvious that the global data distribution is not uniform in every region. The subset covering the North Atlantic flight corridor shows the highest coverage of flights. Therefore, in our study we focus on this region (40N to 60N, -65E to 5E). For the evaluation of the reanalysis data we use the geographic position of the airplane (lat/lon), the time, data quality flags, ambient pressure and temperature (Berkes et al., 2017), relative humidity and water vapour volume mixing ratio. In this study we use the data collected during January 2000 to December 2009. For further information regarding the IAGOS project the reader is referred to the project's website www.iagos.org and the references therein.

## 2.2 ERA-Interim

For this study the ERA-Interim reanalysis data set from the European Centre of Medium-Range Weather Forecasts (ECMWF) is used (Dee et al., 2011). The spectral resolution of the underlying IFS model from 2006 is T255, which calculates to a horizontal resolution of about $80\,\mathrm{km}$ in the mid-latitudes. The vertical dimension is separated into 60 levels reaching from the surface up to a pressure level of $0.1\,\mathrm{hPa}$.

For the comparison with the IAGOS measurements the 6-hourly ERA-Interim data were previously converted on a $1°$ horizontally grid and interpolated on pressure levels (Kunz et al., 2014). Then, the fields of temperature, pressure and specific humidity are projected on the aircraft's flight path by linear spatial and temporal interpolation. Finally, the data is available with a temporal resolution of $4\,\mathrm{s}$ along the flight track (Berkes et al., 2017).

The relative humidity with respect to ice was calculated using the approximation by Murphy and Koop (2005). As mentioned already in the introduction, the ERA-Interim data set was obtained by using the IFS model including the so-called Tompkins scheme (Tompkins et al., 2007), which allows for the supersaturation with respect to ice in cloud-free regions. However, inside of cirrus clouds an occurring supersaturation is adjusted down to $100\,\%$. In the following we use ERA to label the ERA-Interim data.

## 2.3 Methodology

Aircraft based measurements of atmospheric state variables and chemical composition usually refer to the aircraft flight altitude or pressure level, respectively. In the present work the humidity data will be separated relative to the tropopause height in order to study the humidity in the tropopause region. We use the dynamical tropopause, which is defined by a sharp gradient in the potential vorticity (PV). The here used value to define the tropopause is PV = 2 PVU (1 PVU = $10^6\,\mathrm{K\,m^2\,kg^{-1}\,s^{-1}}$, standard potential vorticity unit) (Holton, 2005). A former study (Neis, 2017) showed that the choice of the tropopause definition can have an important impact on the interpretation of the results. However, in this study we want to compare two data sets. Therefore, the definition of the tropopause plays only a minor role. The vertical data will be distributed into three main layers: upper troposphere (UT), tropopause layer (TP), and lower stratosphere (LS) in accordance to Thouret et al. (2006). Furthermore, UT and LS are each separated into three subclasses. The width of the sublayers consider the average difference between ozone and thermal tropopause of $780\,\mathrm{m}$ ($30\,\mathrm{hPa}$ at this altitude) (Bethan et al., 1996). The resulting seven $30\,\mathrm{hPa}$ thick bins separate the aircraft pressure relative to the tropopause pressure and are summarized in Table 1.

| Region | Shortname | $p_{\mathrm{ap}} - p_{\mathrm{tph}}$ [hPa] | number of measurements |
|---|---|---|---|
| | LS3 | -90 | 3 203 483 |
| Lowermost stratosphere | LS2 | -60 | 4 237 245 |
| | LS 1 | -30 | 5 268 138 |
| Tropopause layer | TL | 0 | 5 643 057 |
| | UT1 | +30 | 4 649 883 |
| Uppermost troposphere | UT2 | +60 | 2 647 935 |
| | UT3 | +90 | 909 120 |

**Table 1.** The data set is distributed into three main layers: the upper troposphere, tropopause layer, and lowermost stratosphere. The outer layers are additionally subdivided into three sublayers. The distribution criterion is the pressure difference between aircraft pressure $p_{\mathrm{ac}}$ and the tropopause pressure $p_{\mathrm{tph}}$ with the range of $\pm 15\,\mathrm{hPa}$. Additionally, for every flight layer the number of IAGOS measurements between 2000 and 2009 are presented.

Before the distribution of the data several filters are applied. First, the measured data must lie within the geographic region from $40\,^\circ$ to $60\,^\circ$ North and $-65\,^\circ$ to $5\,^\circ$ East. All data must be collected in a height above $350\,\mathrm{hPa}$ and within a temperature range between $233\,\mathrm{K}$ and $200\,\mathrm{K}$ corresponding to the threshold of homogeneous freezing (Heymsfield and Sabin, 1989) and calibration limit of the humidity sensor (Neis, 2017), respectively. Additionally, the relative humidity with respect to liquid water must be below $100\,\%$ and several measurement quality flags must be fulfilled. The above mentioned criteria are applied to both data sets and is the starting point of the comparison.

## 3   Comparison between IAGOS measurements and ERA-Interim

The aim of this section is to compare and quantify the difference of relative humidity with respect to ice ($\mathrm{RH_i}$) between the in-situ measurements provided by IAGOS and the reanalysis data of ERA-Interim.

### 3.1   Temperature

According to Equation 1 the relative humidity depends on temperature and available water vapour. In a first step, a comparison between the temperature measurements and the reanalyses data for all seven height layers is conducted. Figure 2 shows the vertical profile of the temperature for IAGOS and ERA in the form of a Box-and-Whisker plot. The boxes are bounded by the $25\,\%$ and $75\,\%$ percentile, while the median is marked with a black vertical line. The whiskers are defined as $1.5 \cdot \mathrm{IQ}$ with IQ the interquartile range, being equal the distance between the $25\,\%$ and $75\,\%$. Outliers, values exceeding the whiskers, are marked as black circles. As expected by definition, the tropopause layer shows the coldest mean temperatures with $216.1\,\mathrm{K}$

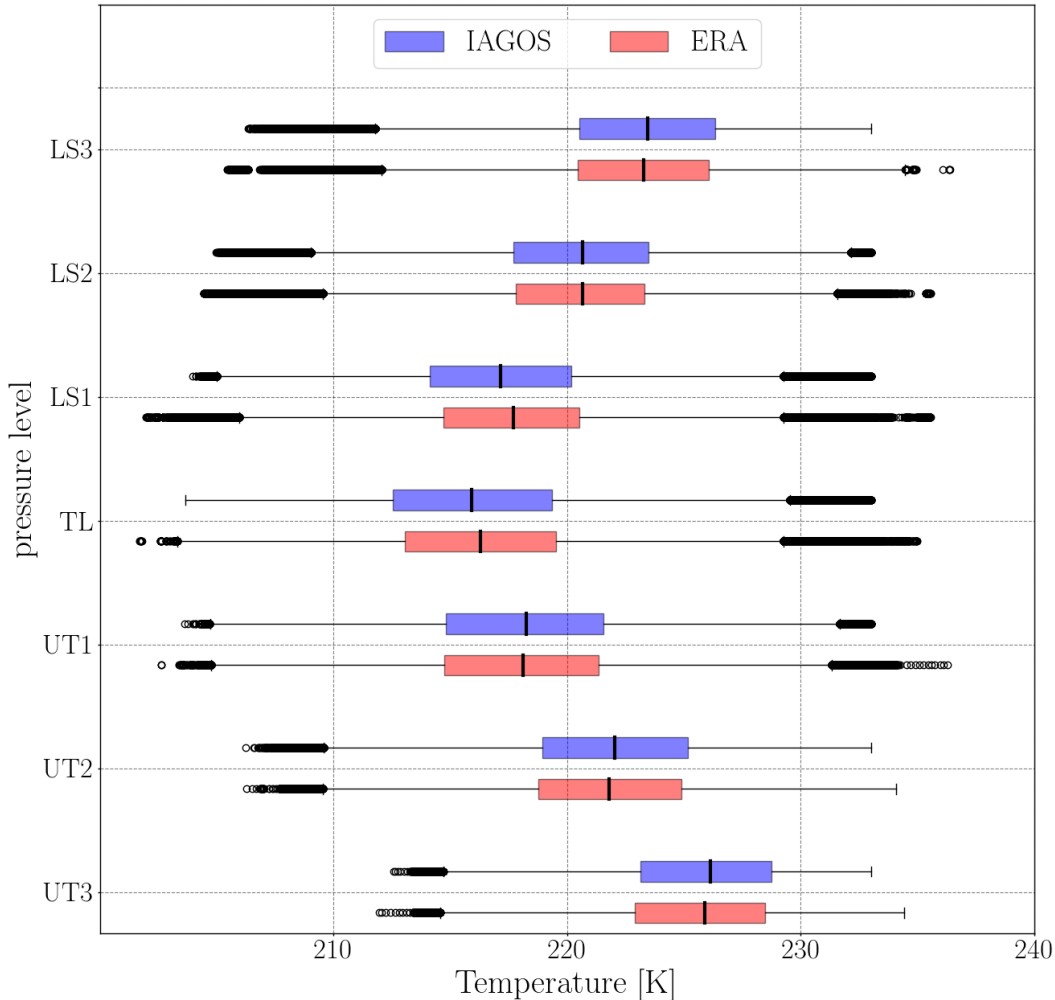

**Figure 2.** Vertical profile of temperature [K] for IAGOS (blue) and ERA (red).

and 216.5 K for IAGOS and ERA, respectively. The warmest mean and median temperatures are visible for the lowest layer UT3 with about 226 K in both cases. Overall, the comparison between IAGOS and ERA shows a very good agreement as far as the statistical values are concerned (see Table 2). This is supported by the equally sized boxes in Fig. 2, pointing to a similar variability of the data set.

| Layer | Median temperature [K] | | Mean temperature [K] | | Standard deviation [K] | |
|---|---|---|---|---|---|---|
| | IAGOS | ERA | IAGOS | ERA | IAGOS | ERA |
| LS3 | 223.5 | 223.3 | 223.3 | 223.2 | 4.2 | 4.1 |
| LS2 | 220.7 | 220.7 | 220.6 | 220.5 | 4.2 | 4.0 |
| LS1 | 217.2 | 217.7 | 217.1 | 217.6 | 4.4 | 4.2 |
| TL | 215.9 | 216.3 | 216.0 | 216.4 | 4.8 | 4.6 |
| UT1 | 218.2 | 218.1 | 218.2 | 218.0 | 4.8 | 4.7 |
| UT2 | 222.0 | 221.8 | 222.0 | 221.8 | 4.4 | 4.3 |
| UT3 | 226.2 | 225.9 | 225.9 | 225.6 | 3.8 | 3.8 |

**Table 2.** Median, mean and standard deviation of temperature [K] for IAGOS and ERA data. The comparison shows a very good agreement.

| Layer | Median VMR [ppmv] | | Mean VMR [ppmv] | | Standard deviation [ppmv] | |
|---|---|---|---|---|---|---|
| | IAGOS | ERA | IAGOS | ERA | IAGOS | ERA |
| LS3 | 21 | 18 | 24 | 22 | 17.2 | 14.7 |
| LS2 | 25 | 26 | 29 | 30 | 19.9 | 17.8 |
| LS1 | 36 | 38 | 42 | 43 | 27.0 | 23.0 |
| TL | 52 | 51 | 61 | 58 | 37.9 | 30.9 |
| UT1 | 75 | 71 | 87 | 80 | 52.7 | 44.4 |
| UT2 | 111 | 106 | 126 | 118 | 71.3 | 62.4 |
| UT3 | 155 | 151 | 174 | 164 | 94.0 | 81.5 |

**Table 3.** Median, mean and standard deviation of water vapour volume mixing ratio [ppmv] for IAGOS and ERA data. The comparsion shows a very good agreement.

## 3.2  Water vapour

The vertical structure of the water vapour volume mixing ratio is presented in Fig. 3. Here, a clear dependence of the distribution with height is visible. The lowest mean and median values of the water vapour volume mixing ratio are observed, as expected, in the uppermost layer LS3. In contrast, the lowermost level UT3 shows the largest mean values of the water vapour mixing ratio. The comparison between IAGOS and ERA shows a good overall agreement (see Table 3). For the upper layers in both data sets starting from the tropopause level (TL) up to LS3 the variability, i.e. the size of the boxes, is significantly lower compared to the tropospheric layers (UT1 to UT3). Overall it can be stated that the variability is increasing with decreasing height. It is notable that the outliers in all layers of IAGOS reach clearly higher mixing ratios than ERA. This has significant effects on the relative humidity with respect to ice, as will be shown later. In summary, the reanalysis data is in good agreement with the vertical distribution of the IAGOS data. However, IAGOS shows a larger variability and stronger extreme values.

| Layer | Median RHi [%] | | Mean RHi [%] | | Standard deviation RHi [%] | |
|---|---|---|---|---|---|---|
| | IAGOS | ERA | IAGOS | ERA | IAGOS | ERA |
| LS3 | 12 | 11 | 15 | 14 | 12 | 10 |
| LS2 | 20 | 22 | 25 | 25 | 17 | 15 |
| LS1 | 47 | 46 | 53 | 49 | 29 | 21 |
| TL | 84 | 77 | 82 | 75 | 29 | 22 |
| UT1 | 93 | 90 | 89 | 83 | 27 | 21 |
| UT2 | 89 | 87 | 85 | 81 | 28 | 22 |
| UT3 | 82 | 81 | 79 | 76 | 30 | 23 |

**Table 4.** Median, mean and standard deviation of RHi [%] for IAGOS and ERA. The comparison shows a good agreement.

Dyroff et al. (2015) reported a moist bias comparing CARIBIC measurements with ECMWF analyses and forecasts. Since ERA-Interim is based on ECMWF analyses one would expect also a moist bias in the lower stratosphere. In contrast to the capacitive sensor used for IAGOS, CARIBIC uses a combination of a frost point hygrometer and a photo-acoustic hygrometer, which shows a better precision and uncertainty for very low water vapour volume mixing ratios. Therefore, the uncertainties in volume mixing ratios of $H_2O$ are large in the lower stratosphere and may explain the more wet stratospheric values compared to CARIBIC. For more information also see Petzold et al. (2019).

### 3.3 Relative humidity with respect to ice

Since cloud formation is governed by the relative humidity rather than the water vapour mixing ratio the relative humidity w.r.t. ice $RH_i$ is now investigated. As mentioned in the introduction, $RH_i$ depends on both temperature and available water vapour. Hence, relative humidity is a convolution of both variables. In Fig. 4 the vertical structure of $RH_i$ is depicted. The overall results show two different regimes. In the troposphere up to the tropopause layer the statistics cover the whole range of possible saturation values. $50\%$ of the data, indicated by the boxes, of each layer from UT3 to TL are situated between $50\%$ and $100\%$ $RH_i$. The highest median values are found in the layer UT1 for both data sets. In the tropopause layer still a significant amount of the data are exceeding values of $RH_i > 100\%$, both in the in-situ data as well as in the ERA data set. However, the whisker in Fig. 4 indicate that ERA has less data points with a higher supersaturation compared to IAGOS. In the stratospheric layers the median of the $RH_i$ values is decreasing strongly. However, ice supersaturation is still possible, especially in the lowest stratospheric layer LS1 (Müller et al., 2015). The statistics of ISSRs in higher levels provided by the ERA data set shows clearly less occurrence of this feature. Since ISSRs are an important factor for the formation and lifetime of contrail cirrus (Kärcher, 2018), a good model representation of the abundance of ice supersaturation in this region is important for an adequate description of the Earth's radiative budget.

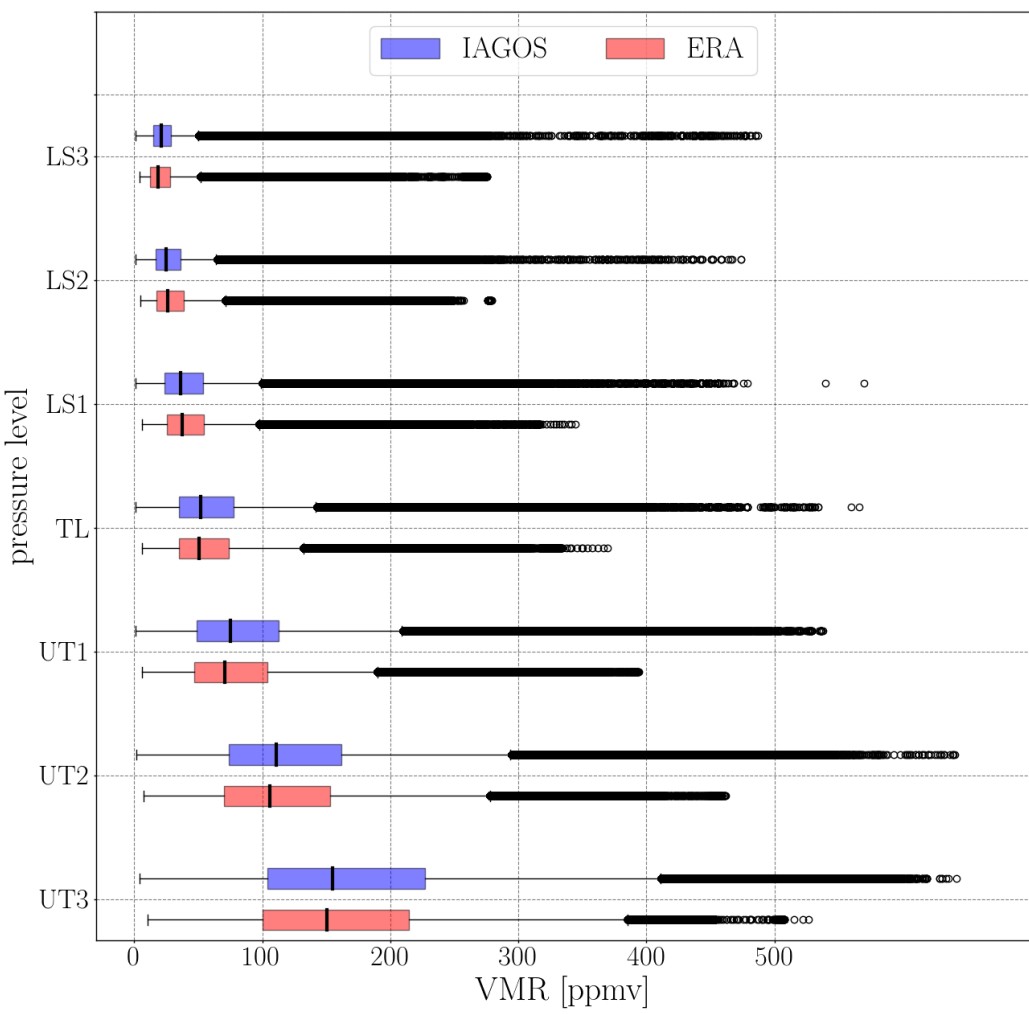

**Figure 3.** Vertical profile of $H_2O$ volume mixing ratio [ppmv] for IAGOS (blue) and ERA (red).

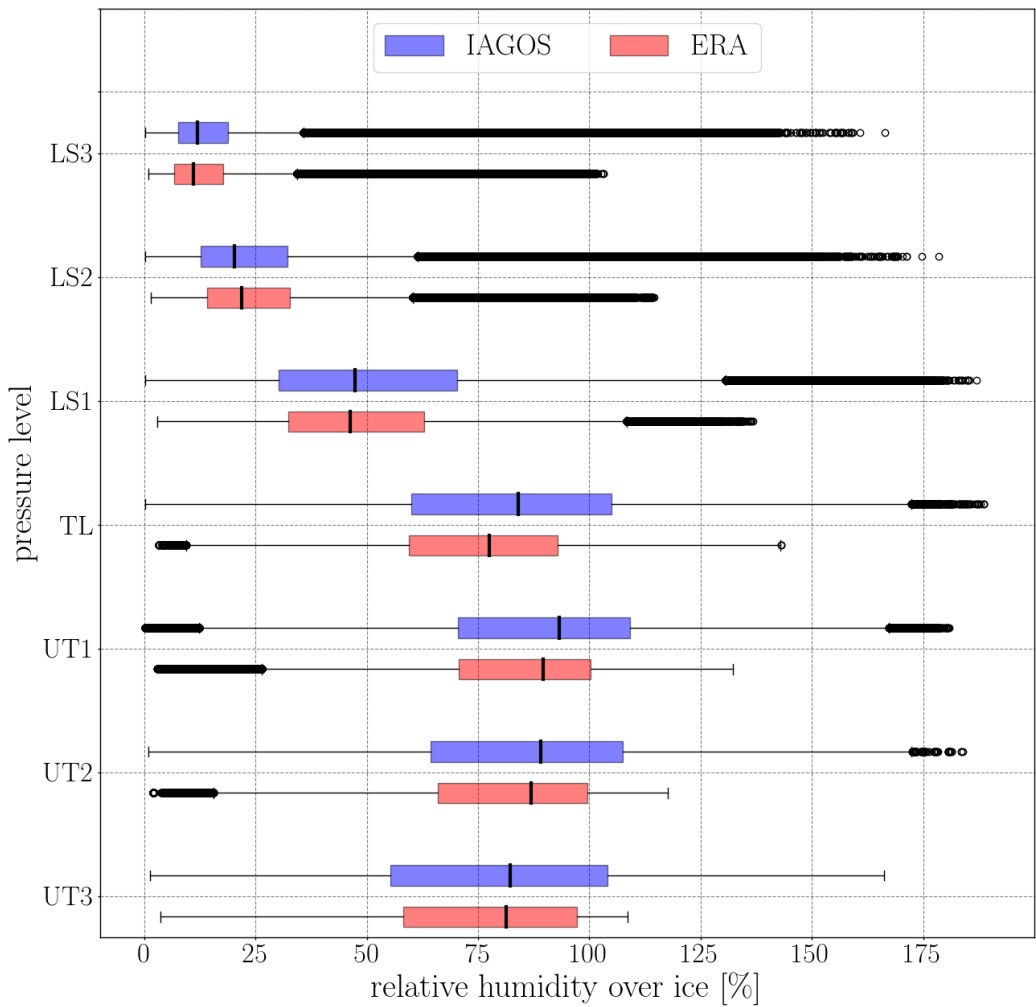

**Figure 4.** Vertical profile of relative humidity with respect to ice (RHi) [%] for IAGOS (blue) and ERA (red).

It is obvious that ERA and IAGOS show a good agreement for situations below ice supersaturation. However, in this study we focus on the situation where ice supersaturation exists. For a more distinct look on the occurrence of ISSR we illustrate the statistic evaluation with cumulative probability in Figure 5. Each layer is depicted with its own color. It is clearly visible that ERA and IAGOS behave differently for $RH_i > 100\,\%$, especially for the tropospheric layers. The ERA distributions of the latter layers snap off as soon as they reach ice supersaturation. As mentioned before, the IFS-model allows the existence of ice

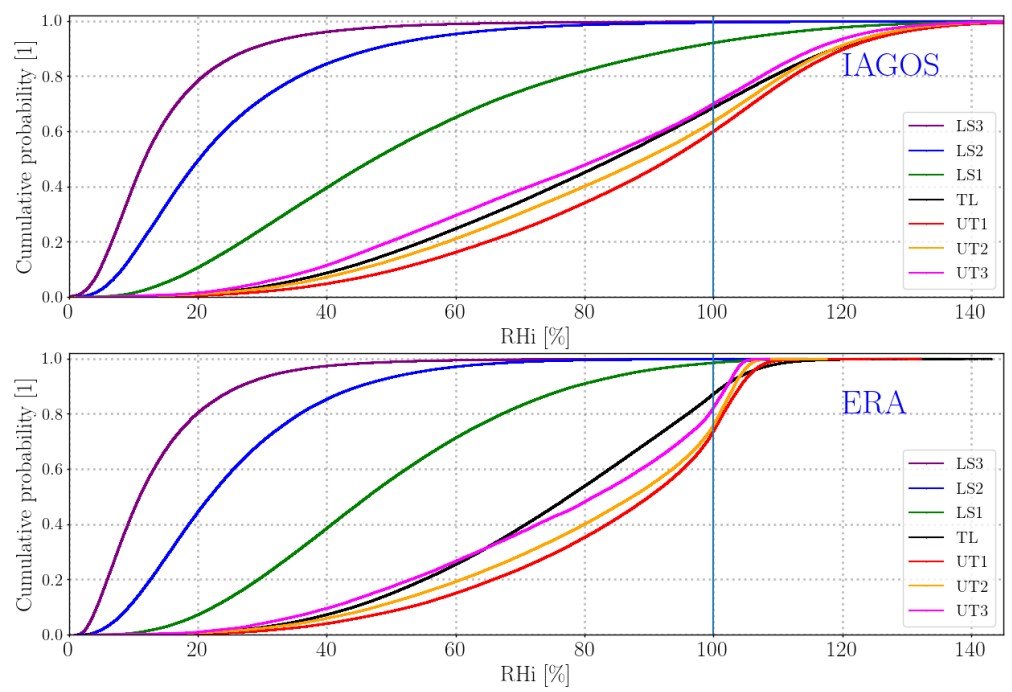

**Figure 5.** Cumulative distribution of $RH_i$ for every height level in the IAGOS (top) and ERA (bottom) data set. The vertical blue line denotes saturation w.r.t. ice.

supersaturation, but only in cloud free conditions. As soon as ice clouds are present in the models grid cell the supersaturation is adjusted to $RH_i = 100\%$. Unfortunately, the IAGOS data set of the investigated time frame cannot distinguish between cloudy and non-cloudy areas. [1] Nevertheless, also in ice clouds ice supersaturation is present (Krämer et al., 2016). Therefore, the behaviour of the cumulative distributions for ERA, especially in the layers from UT3 to TL, might be due to an untimely

5    formation of ice clouds in the underlying IFS model, which adjusts the ice supersaturation too early.

Another way to compare the representation of the water vapour is the fraction of ISSRs. Figure 6 presents the vertical profile of the ISSR fraction. The ISSR fraction in this study is defined as the number of data points within a layer with $RH_i \geq 100\%$ divided by the total number of data points in that layer above the defined North Atlantic region. It is clearly visible that the measurements by IAGOS show a higher fraction of ISSR. Only for the two uppermost layers the fraction of both data sets are

10   of comparable magnitude. Here, the very dry conditions produce only in very few cases supersaturation. The largest difference between measurement and reanalysis data occur in the tropopause layer and the flanking UT1 layer, where a high percentage of ice clouds can be expected. IAGOS shows in the latter layer an ISSR fraction of up to $40\%$. Since ISSR are a prerequisite

---

[1]Nowadays, the IAGOS setup includes an optical sensor for registration of clouds on the flight path.

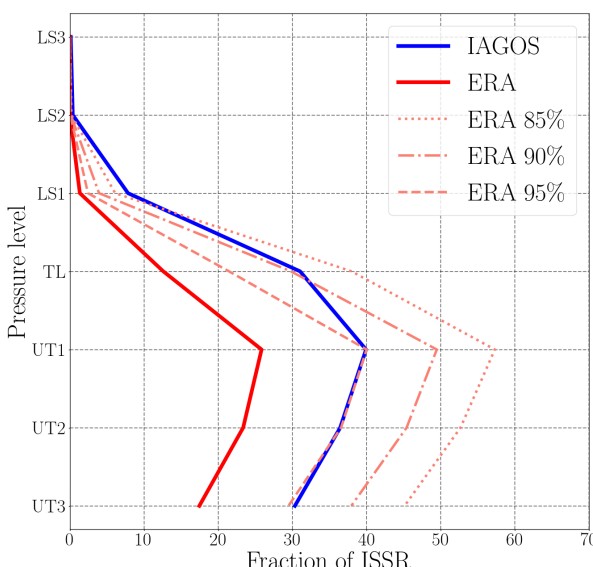

**Figure 6.** Vertical profile of the fraction of ice supersaturated regions for IAGOS (blue) and ERA (red). Different definitions of *ice supersaturation* in ERA are shown in light red. The dotted line represents the fraction of $\mathrm{RH}_{i_{ERA}} \geq 85\%$ , the dashed-dotted line depicts the fraction of $\mathrm{RH}_{i_{ERA}} \geq 90\%$ and the dashed lines stands for the $\mathrm{RH}_{i_{ERA}} \geq 95\%$.

for the formation of in-situ cirrus clouds, a misrepresentation of the feature can lead to great deviations in the local radiative budget and can build up to large errors in the local dynamics.

However, sensitivity studies were conducted in order to see if ERA can reproduce the ISSR fraction of IAGOS using a lower threshold than $\mathrm{RH}_i = 100\%$. For each layer a different $\mathrm{RH}_{i_{ERA}}$ threshold for the ERA data set leads to the best agreement with

5 IAGOS. The best comparison between IAGOS and ERA in the lower stratosphere is found when *ice supersaturation* in ERA is defined as $\mathrm{RH}_{i_{ERA}} \geq 85\%$ (dotted red line in Fig. 6). For the tropopause layer the best agreement is found for $\mathrm{RH}_{i_{ERA}} \geq 90\%$ in ERA (dash-dotted line), while the upper troposphere shows an almost perfect match for $\mathrm{RH}_{i_{ERA}} \geq 95\%$ in ERA (dashed line). This shows that the agreement between measurement and reanalysis is decreasing with increasing height. One reason might be the data assimilation of measurements into the reanalysis data set. The operational radiosondes using capacitive

sensors (usually RS80 and RS92 with Humicap sensor by Vaisala) show large deviations for temperatures below $-40\,^{\circ}\mathrm{C}$ or very low absolute humidities (Spichtinger et al., 2003a). Therefore the response time is increasing in the upper troposphere to the tropopause layer due to the decreasing temperature. Further up in the stratosphere, temperatures are increasing again (Fig. 2). However, the absolute humidities are decreasing with height (Fig. 3), leading to an increased response time.

In contrast to the total fraction over the complete period from 2000 to 2009, Figure 7 shows a time series of the monthly

ISSR fraction. For this comparison the three stratospheric (tropospheric) layers were combined to one layer defined as "lower

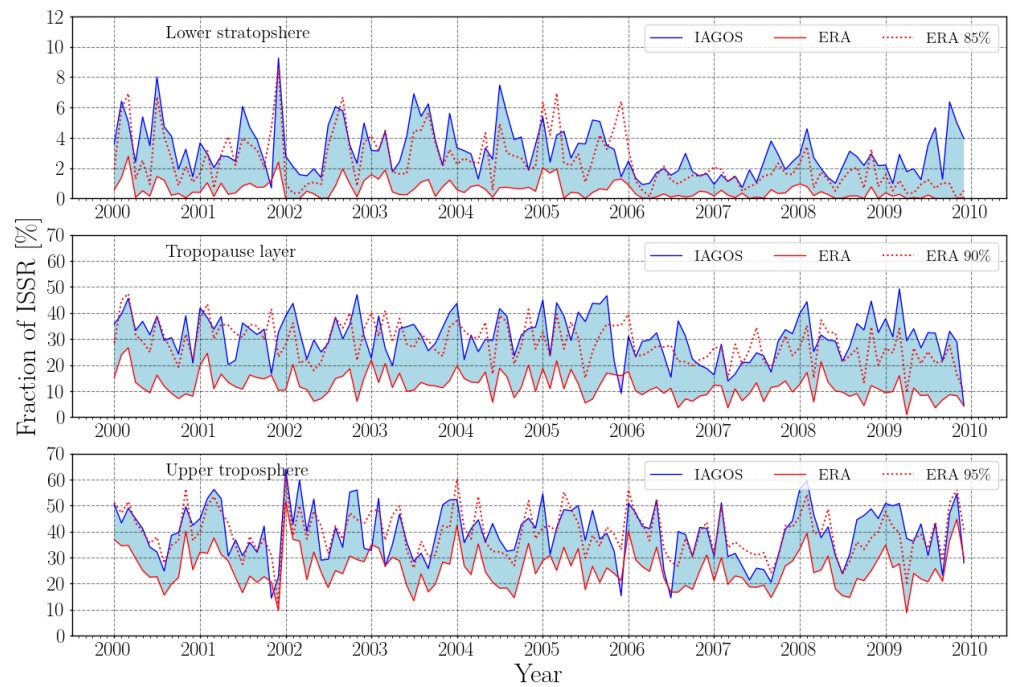

**Figure 7.** Time series of the monthly fraction of ice supersaturated regions for IAGOS (blue) and ERA (red) for the lower stratosphere (LS1 to LS3), the tropopause layer TL and the upper troposphere (UT1 to UT3). The dashed red line represents the ERA data set with the fraction of $RH_i \geq 85\%$ for the lower stratosphere, $RH_i \geq 90\%$ for the tropopause layer and $RH_i \geq 95\%$ for the upper troposphere. Note the different scale for the lower stratosphere.

stratosphere" ("upper troposphere"). The dotted lines in Fig. 7 indicate the fraction of data points for $RH_{i_{ERA}} \geq 85\%$ in the lower stratosphere, for $RH_{i_{ERA}} \geq 90\%$ in the tropopause layer and for $RH_{i_{ERA}} \geq 95\%$ in the upper troposphere, respectively. Note, that the variation of the ISSR fraction is not only affected by meteorology but also due to different data coverage. For instance, in the end of 2001 only few data points are available, which explains the prominent spikes in the lower stratosphere

5 and upper troposphere. Comparing the time series with Fig. 6, it is again visible that ERA is underestimating the fraction of ISSR compared to the measurements from IAGOS. Using a different definition of *ice supersaturation* by reducing the threshold in ERA (85,90 and 95 %) improves the agreement between ERA and IAGOS in each layer. However, the different threshold can also lead to overestimation of the ISSR fraction. While in the lower stratosphere, a threshold of $RH_{i_{ERA}} \geq 85\%$ shows a overall good agreement, in 2005 this threshold would lead to an overestimation of the ISSR fraction of up to $300\%$ (December 2005).

10 It is also noteworthy that, although Fig. 6 shows an almost perfect agreement between IAGOS and the modified threshold of $95\%$ in the upper troposphere, large deviations between IAGOS and ERA can be found as well, for example in 2004.

As a side note, the reader is referred to the companion study by Petzold et al. (2019), where also a trend analysis using the IAGOS data was conducted, which led to the conclusion that no significant trends in ISSR occurrence can be observed. However, they do find a correlation between the North Atlantic oscillation (NAO) and ISSR occurrence.

## 4   Horizontal scales of ice supersaturated regions in IAGOS and ERA-Interim

Although ISSRs are three-dimensional fields the measurements only provide one-dimensional pathlengths. Therefore, only these pathlengths can be used to characterize the size of ice supersaturated regions. We are also interested in the seasonal cycle and height dependence of ISSR pathlengths. Also, the distance between two neighbouring ISSRs are of interest. Additionally, these properties are compared to the reanalysis data and previous studies.

Former studies (Diao et al., 2014; Spichtinger and Leschner, 2016) showed that the horizontal pathlengths can reach from the very small scale in the order of hundred meters to up to $1000\,\mathrm{km}$. Here, small scale variability will lead to very short ISSRs, while large scale features will produce large ISSR pathlenghts.

Figure 8 presents an exemplary flight from Atlanta (USA) to Frankfurt (Germany) on March 7th, 2009. Blue shows the high resolution data available from the IAGOS data base. Red marks the ERA data obtained from the given flight track and the black line shows the pressure level of the airplane. Shaded areas are blue for ice supersaturation in the IAGOS data set and red for ERA. In the upper part of Fig. 8 a satellite image from March 7th, 2009 can be seen. Note, that the flight landed at 7:52 UTC, while the satellite image is taken at 12 UTC, therefore a shift between measurements and image has to be kept in mind. A low pressure system, located between the British Isles and Iceland is dominating the weather over the North Atlantic. The warm conveyor belt (WCB), an ascending airflow from the boundary layer to the upper tropopause (Spichtinger et al., 2005), with its cloud band is clearly visible reaching from Ireland to the south-west. Between a flown distance of $5000$ to $6000\,\mathrm{km}$ an increase in the relative humidity with respect to ice is shown, both in IAGOS and ERA. This is in agreement with other studies, where cirrus clouds and ice supersaturation are found in the ascending air masses of the warm conveyor belt (Spichtinger and Leschner, 2016). Behind the cold front a region of post frontal showers is visible. This is the region of the dry intrusion (Browning, 1997), where stratospheric dry air can be expected at the cruising altitude of the airplane. This can be clearly seen by comparing the height of the airplane (black) and the height of the dynamical tropopause (green) between a flown distance of $4000$ and $5000\,\mathrm{km}$. The tropopause decreases down to a height of $500\,\mathrm{hPa}$ and $\mathrm{RH}_i$ reaches very low values.

The overall agreement between measurement and reanalysis is quite good, when focusing on the patterns of large scale variability. It is obvious that ERA, due to the coarse model resolution, is not able to reproduce the very small fluctuations as seen in the IAGOS data. While the first ISSR after start is reproduced by ERA at the right location, the ISSR at a flown distance of about $6000\,\mathrm{km}$ is misplaced compared to IAGOS. It can be also seen that the pathlength of the first ISSR is significantly larger in ERA. Note, that for this study only ISSRs within the North Atlantic region of this specific flight are taken into account.

In the following we will focus on the length of ISSRs. Therefore, we start with a statistical comparison of the pathlengths between IAGOS and ERA. Here, the influence of the spatial resolution on the statistics of ISSR pathlengths is investigated.

After that, we use the unique IAGOS data set to study in more detail the dependence of ISSR pathlengths on the season and atmospheric layer, i. e. the differences between the lower stratosphere, tropopause and upper troposphere.

## 4.1 Statistic investigation of ISSR pathlengths

It is obvious that the smallest possible pathlength of the ERA data set is limited by the models spatial resolution. However,
by calculating a moving average of the data close to the ERA resolution a comparison is possible. Additionally, we want to answer the question, which resolution is necessary in order to receive a helpful representation of the horizontal scales of ISSRs. A comprehensive representation of ISSRs in atmospheric models enables further investigations on the linkage between small-scale variability and large-scale features.

For a reasonable comparison of ISSR pathlengths between IAGOS and ERA we only evaluate ISSRs on a constant flight
level. This means that an ISSR path is omitted, if the pressure level of the plane is changed by more than $50\,\mathrm{Pa}$. Figure 9 shows the histograms for the pathlengths of IAGOS, ERA and running means of $2, 5, 10, 25, 50$ and $100\,\mathrm{km}$ of the IAGOS data. On the right side, a cumulative distribution is shown. As it is expected from the different underlying resolutions of both data sets, IAGOS (black lines in Fig. 9 shows much more small pathlengths compared to ERA (red lines). On the other side, ERA shows more very large pathlengths larger than $100\mathrm{km}$. This is especially impressive by looking on the cumulative distribution.
Here, the different character of both data sets is most prominent. Comparing the median values given in Table 5 of the original IAGOS data and ERA reveal the big difference. While 50% of the ISSR pathlengths in IAGOS are smaller than $3\mathrm{km}$, the same threshold for the ERA data set is located at $156\,\mathrm{km}$. It has to be noted that due to the lower spatial resolution of the ERA data set only 6283 ISSRs are found compared to $81295$ for IAGOS.

Previous studies also investigated pathlengths of ISSRs. Diao et al. (2014), hereafter D14, investigated ISSR from very high
resolution measurements $(1\,\mathrm{Hz} \approx 230\,\mathrm{m})$ using data from several flight campaigns. They report a mean ISSR pathlength of $3.5\mathrm{km}$, while the median values is about $0.7\mathrm{km}$ with a total number of 1542 ISSRs. These values are about one order of magnitude lower compared to our investigations. However, this data includes measurements from $87\,\mathrm{N}$ to $67\,\mathrm{S}$ and therefore includes tropical conditions.

Spichtinger and Leschner (2016), hereafter SL16, used MOZAIC data from 1995 to 1999 with an original resolution of
about $14\,\mathrm{km}$ and artificially decreased the resolution to $100\,\mathrm{km}$. They report a mean pathlength of $122\,\mathrm{km}$ and a median value of $55\,\mathrm{km}$ for the original data, which are close to our results of IAGOS with a running mean of $10\,\mathrm{km}$. The coarse resolution of SL16 has a mean pathlength of $247\,\mathrm{km}$ and a median value of $149\,\mathrm{km}$. These values are between our results for the running means of $50\,\mathrm{km}$ and $100\,\mathrm{km}$ of the IAGOS data. Additionally, the SL16 coarse data are in very good agreement with the mean and median of the ERA-Interim data.

The comparison of the mean and median values with D14 and SL16 showed, that the results differ due to the underlying resolution of the measurement data. For every scale, a different conclusion can be drawn. Therefore, the question arises, which resolution is feasible to describe the ice supersaturation in a sufficient realistic way for a certain scale. Already with a

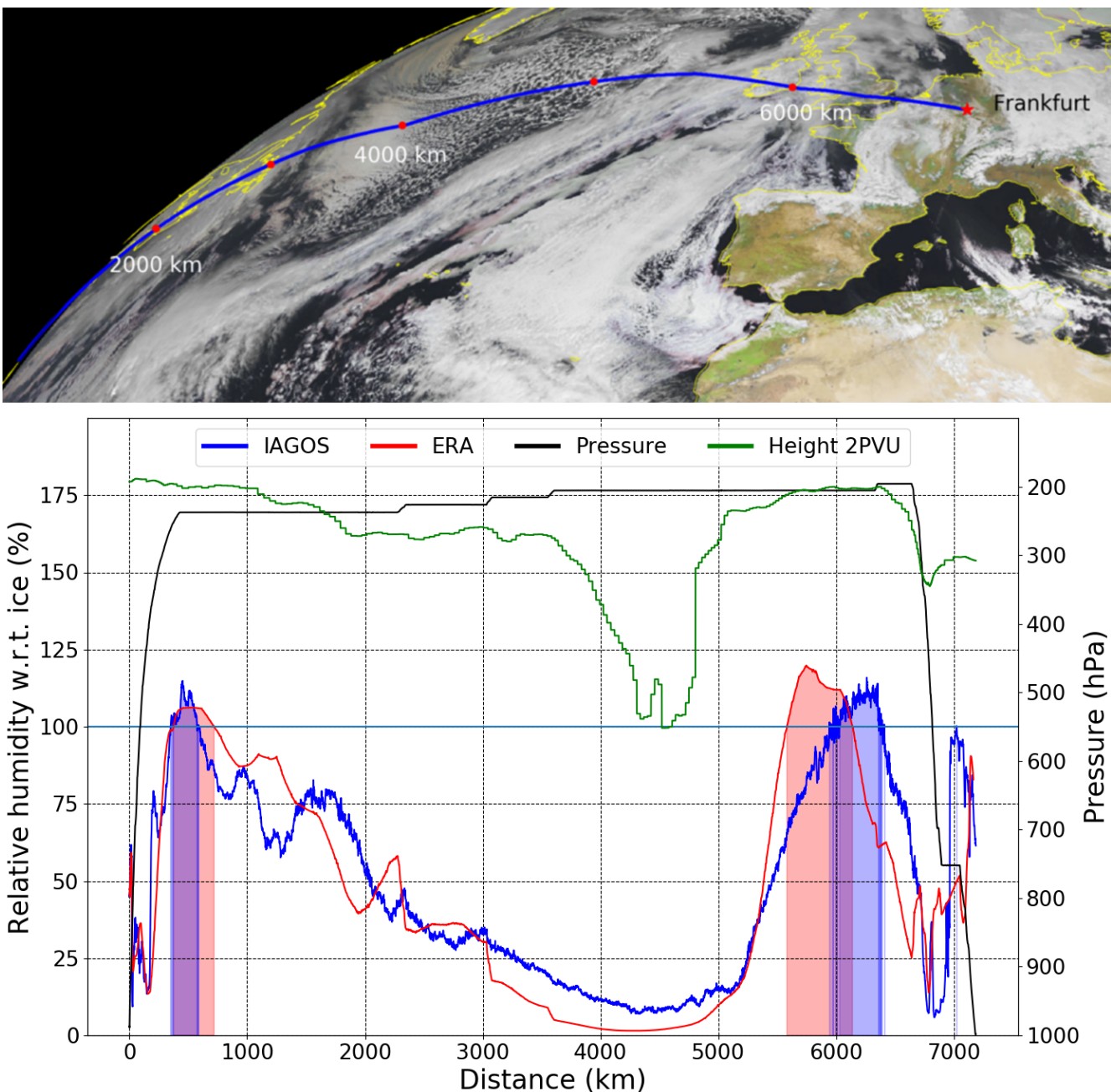

**Figure 8.** Example of typical flight from Atlanta (USA) to Frankfurt (Germany) on March 7th, 2009. The upper part shows the synoptic situation over the North Atlantic on that day including the flight path. For better orientation red dots are placed every 1000 km. In the lower part $RH_i$ is shown over the distance of the flight for IAGOS (blue) and ERA (red). Shaded areas denote ice supersaturation. The pressure level of the aircraft is shown in black and the height in hPa of the dynamical tropopause (2PVU) is shown in green. Satellite image: copyright 2009 EUMETSAT.

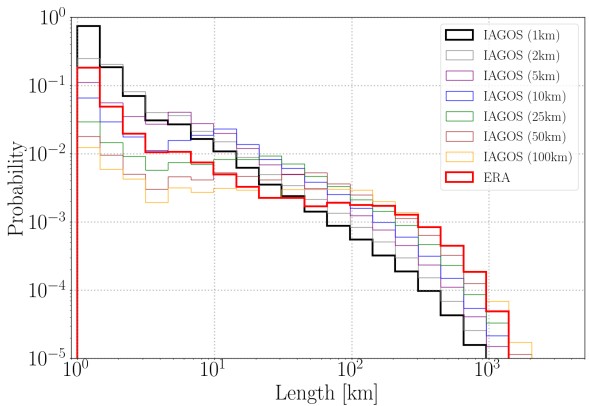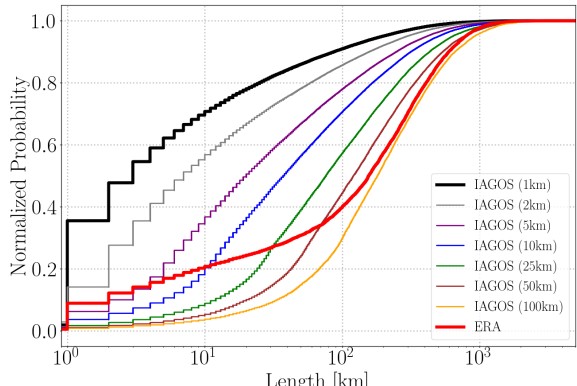

**Figure 9.** Probability of ISSR pathlengths with logarithmic binning (left) and cumulative probability (right). The original IAGOS data is shown in black, while ERA is presented in red. Other colors denote different running means of the original IAGOS data set to mimic different spatial data resolution.

| | IAGOS | ERA-Interim | IAGOS$_{2km}$ | IAGOS$_{5km}$ | IAGOS$_{10km}$ | IAGOS$_{25km}$ | IAGOS$_{50km}$ | IAGOS$_{100km}$ |
|---|---|---|---|---|---|---|---|---|
| Median [km] | 3 | 156 | 7 | 20 | 37 | 76 | 121 | 181 |
| Mean [km] | 38.4 | 243.8 | 58.3 | 89.3 | 118.3 | 170.1 | 223.0 | 287.8 |
| Number of ISSRs | 81 295 | 6 283 | 79 589 | 52 932 | 39 983 | 27 453 | 20 491 | 9 984 |

**Table 5.** Median and mean values and number of ISSR pathlenghts for IAGOS, ERA-Interim and several running means of the original IAGOS data set.

resolution of $2\,km$ the number of small ISSRs is decreasing strongly. The median value increases to $7\,km$, compared to the original resolution. Decreasing the spatial resolution further leads to a decrease in the number of found ISSRs and an increase of the mean and median ISSR pathlength. At a resolution of $10\,km$ the cumulative distribution exhibits a different character compared to the original data. When reaching a running mean of $100\,km$, which is around the spatial resolution of the ERA

5   data set, the number of found ISSRs and median pathlength are in the same order of magnitude as the reanalysis data. When comparing the results of the $100\,km$ running mean of IAGOS with the ERA data set it is also noteworthy that the behaviour for ERA is clearly different for pathlengths smaller than $100\,km$. ERA shows a significantly higher probability for very small ISSRs ($< 10\,km$) than the running mean with $100\,km$. However, the increase in probability to pathlengths of up to $100\,km$ is flatter in ERA. Again, as a reminder, this is the scale of the spatial resolution of ERA.

10   It is clear that the ISSR with pathlengths smaller than $100\,km$ are not represented in the ERA model in the same way as the running mean of IAGOS shows it. This discrepancy on the first look may be small. However, as mentioned before, small ISSRs

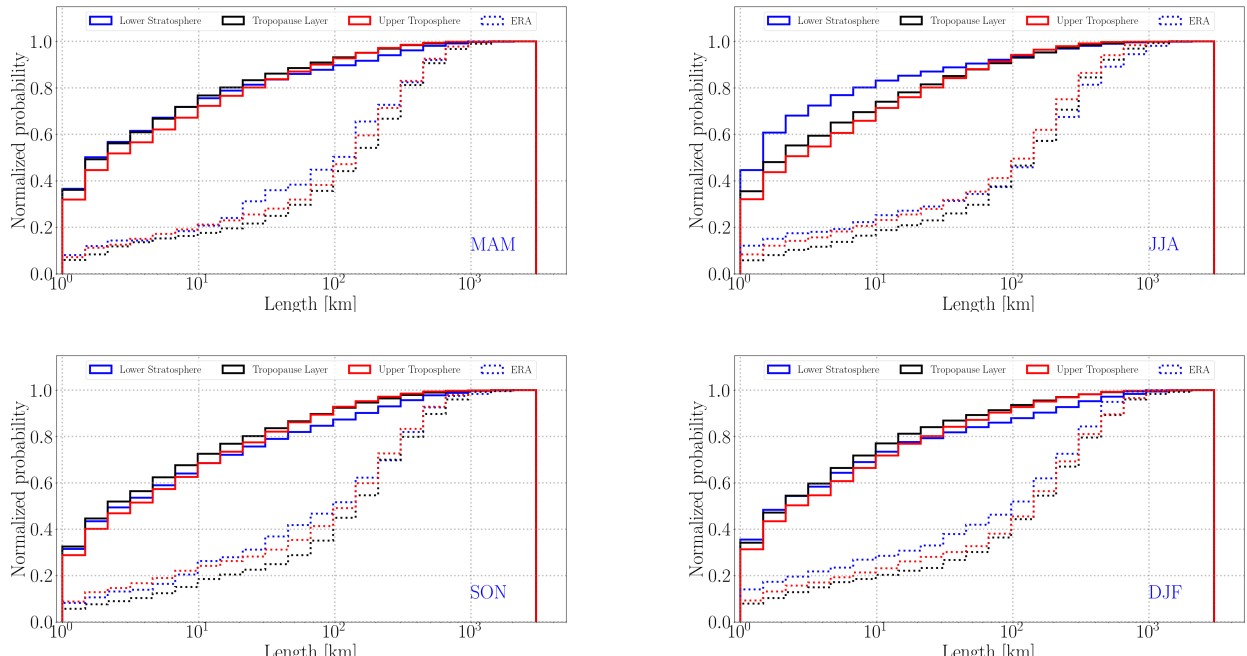

**Figure 10.** Seasonal cycle of ISSRs for lower stratosphere (blue), tropopause (black) and upper tropopshere (red). Dashed lines show results for ERA. Top row presents spring and summer months, bottom row fall and winter months, respectively.

also can have a profound effect as formation region of cirrus clouds. The impact of these smaller ISSRs on the larger scale by changing the dynamics around the tropopause are not clear yet.

## 4.2 Seasonal cycle and height dependence of ISSR pathlengths

The previous results showed the overall statistics of ISSR in the North Atlantic region. Fortunately, the data set allows for a
5  more sophisticated investigation of ISSR properties such as seasonal cycle or height dependence. Therefore, the seasonal cycle for the lower stratosphere, tropopause and upper troposphere is presented in Fig. 10 for IAGOS and ERA. For the sake of clarity, we combine the three upper level to "lower stratosphere" and the three lower layers to "upper troposphere". Starting with spring (MAM), the cumulative distribution shows that for IAGOS in the lower stratosphere and the tropopause region a higher fraction of short ISSRs is present than in the upper troposphere. Up to $75\,\%$ of the ISSRs in the lower stratosphere and
10  the tropopause region are smaller than $10\,\mathrm{km}$. The upper troposphere shows the highest fraction of pathlengths between $10\,\mathrm{km}$ and $100\,\mathrm{km}$. In the lower stratosphere, pathlengths larger than $100\,\mathrm{km}$ are more common on a percentage basis compared to the tropopause and upper troposphere layers.

A clearly different result is visible during the summer months (JJA) in the lower stratosphere. $60\,\%$ of the ISSRs are smaller than $2\,\mathrm{km}$ and over $80\,\%$ smaller than $10\,\mathrm{km}$. The special conditions for the stratosphere in summer are also visible in the

statistical values shown in Table 6. While the mean pathlength in the stratosphere for spring is around $55\,\mathrm{km}$, this value decreases to $35.8\,\mathrm{km}$ in summer, before an increase to approximately $70\,\mathrm{km}$ in fall and winter is observed. It has to be noted that the number of ISSRs in the stratosphere during summer is over two times higher compared to the other seasons.

It is not clear, what kind of process is responsible for the large amount of small ISSRs during summer in the lower strato-
sphere. In principle, relative humidity w.r.t. ice can be altered by a change of temperature, pressure or specific humidity. Processes behind these changes can be especially adiabatic expansion or mixing of different air masses. SL16 investigated the origin of small scale ISSRs and concluded that most of the variation of RHi is due to adiabatic processes, i. e. cooling by expansion. This means that the transport of air masses from different altitudes create small scale variations of RHi in the region from the upper troposphere to the lower stratosphere. However, a deeper investigation of this question requires three-dimensional
data and trajectory analysis, which are not in the scope of the present study but should be conducted in the future.

The results for fall (SON) and winter (DJF) are very similar. The tropopause region shows the highest percentage of ISSRs with a pathlength smaller than $10\,\mathrm{km}$. Pathlengths between $10\,\mathrm{km}$ and $100\,\mathrm{km}$ show the highest fraction in the upper troposphere while the highest percentage of ISSR with sizes larger than $100\,\mathrm{km}$ can be found in the lower stratosphere.

The results for ERA (dotted lines) in spring, and also for the other seasons, are clearly different. As mentioned above, short
ISSRs are underestimated by ERA due to the spatial resolution, which is visible for all seasons. The different resolution leads also to other relations of the pathlengths between different height levels. For instance, the highest fraction of ISSR in spring with a pathlength between $10\,\mathrm{km}$ and $100\,\mathrm{km}$ is found in the lower stratosphere. This behaviour is reversed compared to the IAGOS data set, where the upper troposphere showed the highest fraction. The same observation can be made in winter as well. Not surprisingly, ERA misses also the high fraction of short ISSRs in the stratosphere during the summer season.

In general, the mean pathlength of an ISSR in the IAGOS data set is significantly larger in the stratosphere compared to the upper troposphere and tropopause layer, except for the summer season, which is also shown in Fig. 11. Comparing the mean pathlengths for every season in the IAGOS data makes clear that the strongest seasonal cycle is present in the lower stratosphere with the lowest mean pathlength in summer and the highest values in fall and winter. This might be due to the enhanced storm track activity during fall and winter, where the mass flux (including water vapour) from the troposphere to the
stratosphere is largest (Reutter et al., 2015). Further investigations are needed to answer this question.

The lower two layers also show a seasonal cycle, however with a smaller amplitude and shorter absolute mean values. While the upper troposphere also shows the shortest mean pathlength in the summer, the minimum in the tropopause is found during winter. Note, due to the large number of very small pathlengths in all seasons and all levels and the logarithmic binning of the results, the median values show only a small interseasonal fluctuation. Figure 11 also shows the seasonal cycle of the mean
and median pathlength for the ERA data set. As expected, due to the large resolution of the data set also the mean and median values are significantly larger compared to IAGOS. More importantly, the seasonal cycle of ERA is shifted, especially for the tropopause and lower stratosphere. The maximum mean pathlength in the tropopause can be found in summer for IAGOS, where in the ERA data the minimum pathlength is found. The same holds true for the lower stratosphere, where the minimum for IAGOS is also found in summer, where ERA exhibits the maximum. Additionally, the relative amplitude of the seasonal

| | Upper Troposphere | | | Tropopause | | | Lower Stratosphere | | |
|---|---|---|---|---|---|---|---|---|---|
| Season | N | mean(L) | median(L) | N | mean(L) | median(L) | N | mean(L) | median(L) |
| MAM | 12523 | 36.3 | 3.0 | 10317 | 35.7 | 3.0 | 3995 | 55.5 | 2.0 |
| JJA | 19672 | 34.1 | 3.0 | 13531 | 39.4 | 3.0 | 10924 | 35.8 | 2.0 |
| SON | 19517 | 40.2 | 4.0 | 9602 | 41.0 | 3.0 | 3630 | 69.5 | 4.0 |
| DJF | 13228 | 36.4 | 3.0 | 8268 | 34.5 | 2.0 | 3598 | 67.9 | 2.0 |

**Table 6.** Number N, Mean and median of ISSR pathlenghts L [km] for the upper troposphere, tropopause and lower stratosphere for the seasons MAM, JJA, SON and DJF.

cycle is smaller compared to IAGOS, most prominent in the lower stratosphere. Here, also the spread of the data, indicated by the size of the boxes, shows a large fluctuation in the IAGOS data set, which is not caputred by ERA.

SL16 also investigated the seasonal cycle of ISSRs. All ISSR of the extratropcis (latitude $\geq 30°$) were separated into "tro-
posphere", "stratosphere" and "in between" using the ozone mixing ratio. This chemical definition of the tropopause is close to the thermal definition. However, in our study we use the dynamical definition of the tropopause. This has to be kept in mind for the interpretation of the comparison. As mentioned above the data used in SL16 showed a coarser resolution of about $1\,\mathrm{min}$ (which converts to about $14\,\mathrm{km}$ depending on the true air speed of the plane) in contrast to the much higher $4\,\mathrm{s}$ (about $1\,\mathrm{km}$) resolution in this study. They focused on the large-scale aspects of ISSRs and therefore the original data was analyzed
with a coarse resolution (realized with a running mean of the original data) of about $100\,\mathrm{km}$. The coarser resolution makes a quantitative comparison difficult, since already $60-80\,\%$ of the ISSRs in our study are smaller than $10\,\mathrm{km}$, i.e. below the spatial resolution of the original data in SL16. The maximum of the mean pathlength in SL16 for the stratosphere is found in winter, while the minimum is found in fall. This is in contrast to the finding in the present study, where on the one hand in winter also the maximum pathlengths are found, but the minimum is found in summer. Additionally, during fall about the
same mean pathlength can be found as in winter. The comparison of the seasonal cycle for the tropopause layer between SL16 and the present study reveals also a shift. The maximum mean pathlength in SL16 is found in fall, where our study detect the maximum in summer. The seasonal cycle in the tropospheric layer in SL16 and the present study is in good agreement.

It seems that the difference in the seasonal cycle between SL16 and our study can be, to a large fraction, attributed to the spatial resolution of the data. Figure 11 shows also the seasonal cycle for ERA, where the agreement with SL16 is better. Also,
as mentioned in the discussion of Table 5, the agreement between ERA, the running mean of $50\,\mathrm{km}$ and $100\,\mathrm{km}$ as well as the results of SL16 is very good. Therefore, it is important to have the awareness that the spatial resolution can strongly influence the results of ISSR pathlengths.

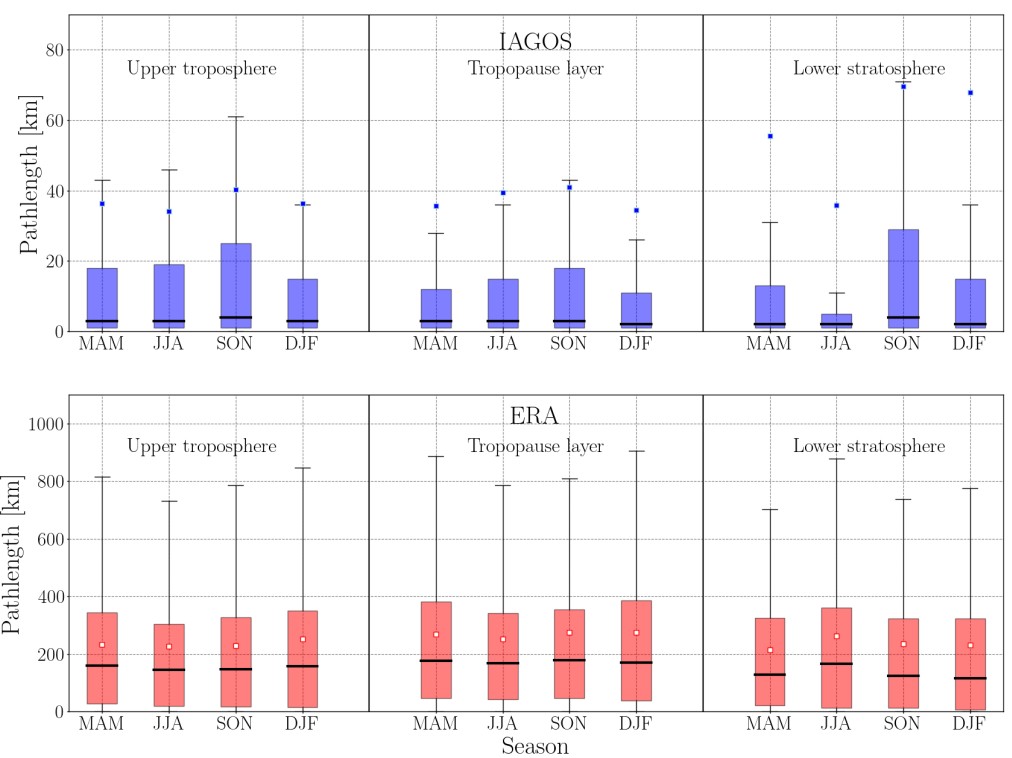

**Figure 11.** Box-and-Whisker plot of the seasonal cycle of mean and median pathlengths of ISSR for different atmospheric layers in the IAGOS data set. Solid blue line shows the lower stratosphere (LS), solid black line the tropopause layer (TL), and solid red line upper troposphere (UT) for the IAGOS data set. Light colors denote the results for the ERA data set. Dashed lines denote the median pathlength, respectively.

| | Upper Troposphere | | | Tropopause | | | Lower Stratosphere | | |
|--------|-------|---------|-----------|-------|---------|-----------|-------|---------|-----------|
| Season | N | mean(D) | median(D) | N | mean(D) | median(D) | N | mean(D) | median(D) |
| MAM | 11784 | 43.8 | 3.0 | 9429 | 50.8 | 2.0 | 3198 | 49.3 | 3.0 |
| JJA | 18367 | 54.7 | 3.0 | 12540 | 60.9 | 3.0 | 9828 | 52.6 | 3.0 |
| SON | 18362 | 55.0 | 3.0 | 8754 | 67.8 | 3.0 | 2789 | 79.1 | 3.0 |
| DJF | 12578 | 35.3 | 2.0 | 7589 | 39.1 | 2.0 | 2822 | 40.3 | 2.0 |

**Table 7.** Mean and median of the distance D [km] between ISSRs in the IAGOS data set for the upper troposphere, tropopause and lower stratosphere for the seasons MAM, JJA, SON and DJF.

## 4.3 Seasonal cycle and height dependence of the distances between ISSRs

Finally, we present the seasonal cycle of the distance between two neighbouring ISSRs. Figure 12 shows the cumulative distributions of the distance between ISSRs for IAGOS and ERA, while in Table 7 the statistical overview for IAGOS is given. Here, a clear seasonal cycle is visible in all three layers, in contrast to the pathlengths of ISSR. For all layer, the maximum of the mean distance is found in fall, with the largest value of $79.1\,\mathrm{km}$ in the lower stratosphere and the smallest distance in the upper troposphere with $55\,\mathrm{km}$. The shortest mean distances are found in winter for all layers with values from $35.3\,\mathrm{km}$ in the upper troposphere to $40.3\,\mathrm{km}$ in the lower stratosphere.

A shorter mean or median distances implies, that ISSRs are closer to each other (D14). Therefore, in winter the conditions favour ISSRs with smaller distances. This also suggests that in the upper troposphere and the tropopause region, where the interseasonal mean pathlength of ISSR does not change strongly, the structure of ISSR is more heterogeneous than in other seasons. For the lower stratosphere one has to keep in mind, that the mean pathlength in winter is, together with fall, significantly larger than in spring or summer. Therefore, not only the distance between ISSR is shortest, but also the pathlength is largest.

In D14, the authors presented a mean distance between ISSRs about $47\,\mathrm{km}$, while the median distance is approximately $1\,\mathrm{km}$. This values are in good agreement with our findings. However, as mentioned before, D14 used global data without the distinction between geographical region or height. SL16 also investigated the distance between ISSRs. The mean and median values are significantly larger than in the present study, which is, again, due to the different spatial resolution. Also, as for the ISSR pathlength, the seasonal cycle is shifted compared to the higher resolution in our work.

The results for ERA (dotted lines in Fig. 12) show again a clearly different picture. The comparison of the height dependence between IAGOS and ERA reveals a reversed and even enhanced dependence as it is shown by the distance of the vertical layers within the ERA data set. For instance in winter, the reversed dependence with height is most prominent. Between $1\,\mathrm{km}$ and $100\,\mathrm{km}$ the lower stratosphere shows the highest fraction of distances, followed by the tropopause layer and the upper troposphere. In ERA, this dependence is vice versa, with the highest fraction in the upper troposphere and the lowest fraction in the lower stratosphere.

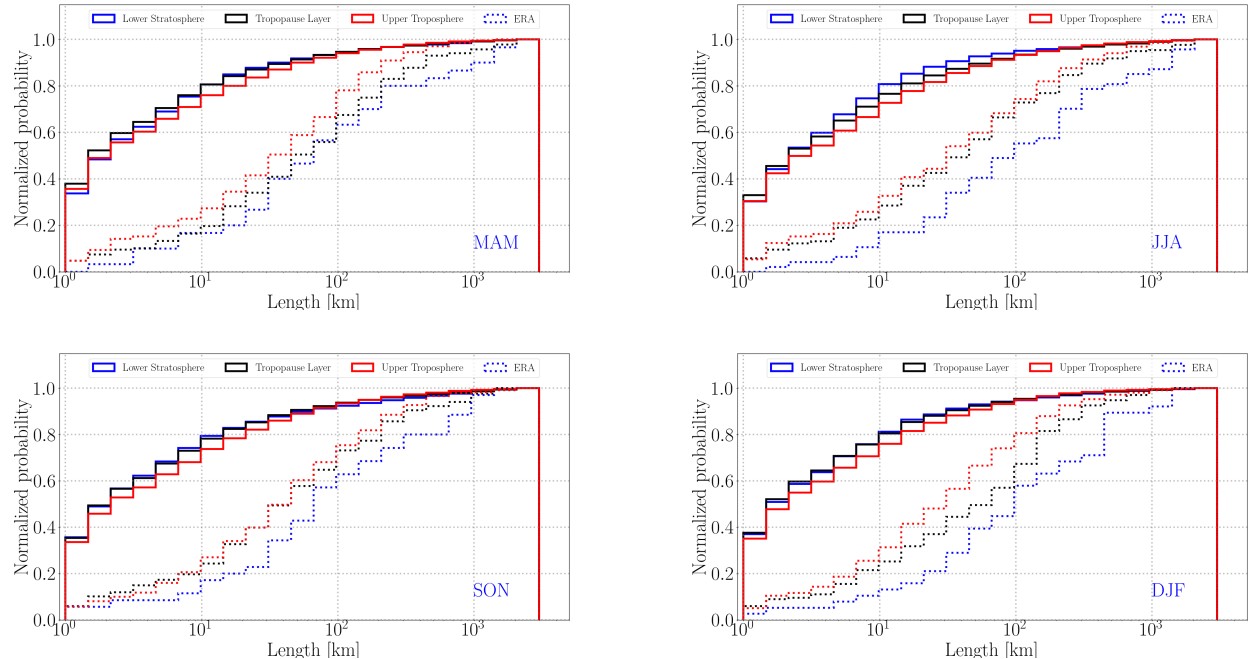

**Figure 12.** Seasonal cycle of the distance between ISSRs for lower stratosphere (blue), tropopause (black) and upper tropopshere (red). Dashed lines show results for ERA. Top row presents spring and summer months, bottom row fall and winter months, respectively.

## 5 Conclusions

This study compares the in-situ measurements of temperature and water vapour and subsequent relative humidity with respect to ice in the UTLS region obtained by IAGOS with the reanalysis data set ERA-Interim from the year 2000 to 2009 over the North Atlantic. Ice supersaturated regions (ISSRs) are of special interest in this investigation due to their abundance and
importance on the local radiation budget when transformed to a cirrus cloud. Additionally, the characteristics of the horizontal scale of ISSRs is investigated, including the seasonal cycle and height dependence. Both data sets are separated according to their relative height compared to the dynamical tropopause (2PVU). The comparison of the temperature shows a good agreement between measurement and reanalysis data. The structure and variability of the vertical temperature distribution is very similar, shown by a very good accordance in median and mean values as well as in the standard deviation. The water vapour
was analyzed using the water vapour mixing ratio. Both data sets show the clear decrease of water vapour with increasing height. In contrast to the measurements ERA shows clearly less variability, indicated by smaller standard deviations in all levels. The convolution of water vapour and temperature leads to the relative humidity with respect to ice, which governs the cloud formation. Both data sets reproduce two different regimes. In the UT layers including the tropopause layer, the statistics cover the whole range of possible saturation values, where most of the data lie between $50\%$ and $100\%\,\mathrm{RH}_i$. However, ERA
deviates from the $\mathrm{RH}_i$ measurements concerning values of larger than $\mathrm{RH_i} = 100\,\%$ by showing less data points and weaker

supersaturations, impressively depicted by the comparison of cumulative distributions. This is an important finding, because it points to a misrepresentation in ERA of ice supersaturation in the UT and tropopause region, which is the formation region of in-situ cirrus. Moving up to the stratospheric layers the $\mathrm{RH}_i$ values, as expected, are much lower in both data sets. Again, ERA shows less and weaker supersaturations through all levels. Since the LS is very dry, supersaturation occurs rarely and therefore

the difference between both data sets is smaller regarding the cumulative distribution. Nevertheless, ERA shows clearly less extreme events. The strong differences between IAGOS and ERA with respect to ISSRs is also shown by the fraction of ISSR. In the UT and the tropopause region the measurements show a significantly larger fraction of ISSR in the measurements compared to ERA.

The comparison of pathlengths of ISSRs shows clearly the different resolutions of the two data sets. It is obvious that the

high-resolution measurements show more small ISSRs than ERA. Only beginning with pathlengths in the order of $100\,\mathrm{km}$ the distribution start to have a similar course. This length scale coincides with the horizontal resolution of the underlying ERA model. Decreasing the resolution of the IAGOS data by running means shows only a good agreement of model and measurement beginning with a running mean of $100\,\mathrm{km}$, which is, as stated, in the order of the model resolution. However, even in the latter case, the structure of the distribution of ISSR smaller than $100\,\mathrm{km}$ is clearly different between IAGOS and

ERA. Therefore, a simple increase of the model resolution seems not sufficient Additionally, the physical processes must be refined carefully for a realistic description of the ice supersaturation in the UTLS region.

The investigation of the horizontal scales of ISSRs provides several results. First, the high resolution data from IAGOS showed a very high percentage of small-scall ISSRs, which is highlighted by median values between $2$ and $4\,\mathrm{km}$ for all heights during all seasons. Up to $80\,\%$ of the ISSRs are smaller than $10\,\mathrm{km}$. The seasonal cycle of the ISSR pathlengths is small

for the upper troposphere and tropopause region, but shows a distinct change for the lower stratosphere during the course of the year. Here, during summer the mean pathlength of an ISSR is lowest with $35.8\,\mathrm{km}$ compared to the maximum during fall with $69.5\,\mathrm{km}$ The comparison with previous studies using different spatial resolutions as well as ERA showed, that the characteristics of the horizontal scale strongly depends on the data structure. Not only the statistical values like mean and median values differ strongly, but also the seasonal cycle is shifted when using different resolutions. Additionally, the sign of

the height dependence is changed when comparing IAGOS with ERA.

The results for the distance between ISSRs shows the same behaviour than the pathlengths. Here, also the spatial resolution of the data has an influence on the resulting seasonal cycle and height dependence. This shows the limitations of one-dimensional data along flight tracks.

The influence of the spatial resolution on the results on ISSR pathlengths was shown in this study. The size, magnitude and

seasonal cycle of ISSRs are closely related to the underlying spatial resolution of the data. Hence, future studies should focus on three-dimensional data from models for further investigations of physical processes regarding ice supersaturation in the UTLS region.

*Author contributions.* PR and PN performed the analyses and both wrote the text; PN and SR were in charge of the instrument setup, calibration, and processing of the measurements; BS combined reanalysis data with measurements; BS and SR checked the manuscript.

*Competing interests.* The authors declare that they have no conflict of interest.

*Acknowledgements.* The authors thank two anonymous reviewer for their very helpful comments and suggestions, which improved the quality of this manuscript. Philipp Reutter and Patrick Neis want to thank Peter Spichtinger, Andreas Petzold, Herman G. J. Smit and Peter Hoor for many helpful discussions.

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
