# Peer review of "Ice supersaturated regions: properties and validation of ERA-Interim Reanalysis with IAGOS in-situ water vapour measurements"

_Atmospheric Chemistry and Physics, 2019_

## Referee Comment (RC1) · Anonymous Referee #1 · 9 Aug 2019

The study of Reutter et al. uses the IAGOS measurement data set to evaluate water vapour and ice saturated regions in ERA-Interim. The manuscript provides meaningful information that should be considered in further investigations of clouds and radiation in the UTLS using ERA-Interim. The manuscript is well and clearly written and the figures are appropriate. The results are meaningful interpreted and understandable although they could be more in-depth at some places. So far, the focus is quite technical because it only describes the differences between the data sets. According to the aims of the journal, studies investigating "chemical and physical processes" are requested. You may add scientific value to your study by further dividing your data set. You could e.g. try if there are differences in your results when you subdivide the data according to season, latitude or elevation of the tropopause. Nevertheless, I recom-

mend the manuscript for publication if modifications are made the following comments are addressed:

Specific questions:

1. I would use a more applicable title such as: Validation of ECMWF ERA-Interim Reanalysis with IAGOS in-situ water vapour measurements in the UTLS region

2. How may data points does your comparison involve (and how many in each level)? Instead of the PDFs in Fig. 1 you could plot the total number of measurements per grid box.

3. I don't see much added value of showing PDFs and box plots of the same data especially because they are not interpreted in detail each. I would prefer the box plots in the paper and to just mention the shape of the PDFs where required. Showing the two data sets in one box plot side by side would make the comparison easier.

4. Fig. 2: Could be the non-Gaussian distribution of temperature values in UT3 be a result of the used tropopause definition?

5. The wave-like structure in Fig. 2 (mentioned on p. 7 l. 3) would be worth investigating to increase the scientific statement of the paper, especially because it is not only visible in UT3, but continues to UT2 and UT1. You could analyse the data of the discrete steps or waves, separately and explore their properties (see scientific suggestions above).

6. subsection 3.2: Dyroff et al. 2015 found a moist bias in ECMWF analyses and forecasts in the lower stratosphere. Please discuss why this bias is not apparent in your data.

7. P. 11 l. 7: The sentence is a bit weird: Do you call ERA-Interim a climate model or do you draw from your results that climate models must show an underestimation of ISSRs as well? Please reconsider/reformulate.
8. Do the negative RHi for IAGOS in the LS in Fig. 6 make sense?

9. How does your statistics of temperature and mixing ratio (Fig. 3, 5) appear inside (and outside) of the ISSRs? Is there a way to relate the underestimation of RHi>100% in ERA to either of these variables?

10. P. 12 l. 6ff and Fig. 8: Does your ISSR fraction mean the (relative) number of ISSRs among all data points? Please clarify.

11. subsection 3.4.1: What are the flight distances of your examples? It would prefer a length axis instead of or in addition to time in Fig. 9.

12. subsection 3.4.1: To add some scientific content it would be very intersting to discuss the weather situation behind the flights, e.g. with a satellite image.

13. As far as I understood the first paragraph of 3.4.2 refers to the comparison of IAGOS (1km) and ERA in Fig. 10? If so, it would help to draw the reader's attention to the respective lines in the figures e.g. (see black and red lines in Fig. 10).

14. P. 14 l. 8: "Already with a resolution of 1km the cumulative distribution exhibits a different character." Different to what? Or do you mean "10km" instead of "1km"?

15. P. 15 l. 1: What exactly do you mean by: "...the behaviour of the ERA data set changes for pathlengths of 100km..."? Please revise.

16. P. 16 l. 8-11: This is part of the motivation and should appear much earlier, in the introduction.

Additional comments:

1. P. 1 l. 23: Remove the "s" from clouds

2. P. 2 l. 25: hyydroxyl?

3. P. 9 l. 14: "that IAGOS data set shows": add "the" before IAGOS or remove "set"

4. P. 11 l. 11: RHi>1: Do you mean RHi>100%?

5. P. 12 l. 9: Remove double "the"

6. P 12 l. 14: Correct "reanaylsis"

7. P. 14 l. 10: Delete one "s" in "median values increases"

8. P. 15 l. 5. Lower case "s" in ISSRS

9. P. 15 l. 26: I would change "Moving up to the stratospheric layers, as expected, the RHi values are" to "...layers the RHi values, as expected, are"

Literature:

Dyroff, C., A. Zahn, E. Christner, R. Forbes, A. M. Tompkins, and P. F. J. van Velthoven, 2015: Comparison of ECMWF analysis and forecast humidity data with CARIBIC upper troposphere and lower stratosphere observations. Quart. J. Roy. Meteor. Soc., 141, 833–844, https://doi.org/10.1002/qj.2400.

---

## Referee Comment (RC2) · Anonymous Referee #2 · 31 Aug 2019

Review of the study entitled 'Comparison of IAGOS in-situ water vapour measurements and ECMWF ERA-Interim Reanalysis data' by Reutter et al.

The study compares 10 years of IAGOS measurements of air temperature, water vapour and relative humidity with ERA-Interim reanalysis data near the tropopause over the North Atlantic Ocean. The analysis focuses on regions substantially saturated with respect to ice, known as ice supersaturated regions, which are regions in which cirrus clouds are formed in the northern mid-latitudes. Comparisons are performed using statistics involving the median, mean and standard deviation, and are illustrated using probability density functions (non-cumulative and cumulative) and box-and-whisker diagrams. The comparisons refer to the period 2000 to 2010. The study is well written and I recommend publication after revision as follows:

[Figure]

Major comments

The study shows the good ability of IAGOS measurements to capture small scale IS-SRs (smaller than 100 km) and at the same time gives credit to the ERA-Interim model to depict large scale ISSRs (larger than 100km). These are important findings which merit publication as far as our knowledge on the detection of ISSRs from different datasets. However, presenting only pdfs and boxplots in a comparison study is not sufficient to justify publication in a journal as such ACP. The authors should make deeper comparisons with their ISS data. For instance:

1) They could perform simple time series analysis for their region (40-60N, 5-65W) and plot the monthly time series of the two ISS datasets from 2000 to 2010, compare the seasonal cycles, and then correlate the two time series after removing the seasonal variability.

2) It has been shown that cirrus cloud variability is significantly affected by the North Atlantic Oscillation during winter (Eleftheratos et al., 2007). The authors could test if a correlation between the deseasonalized ISS data and the NAO index exists.

Technical corrections

P3 l16: correct 'asses' to 'assess'.

P3 l26: correct 'continous' to 'continuous', correct 'greenhous' to 'greenhouse'.

P4 l2: you say '40N-60N, 5-65W' but on p5 l16 you write 'from 40o to 60o North and -65o to 5o East'. Please write the correct coordinates for longitude.

P4 l15: what is the '4s resolution'?

P8 table 3: for the case of TL, the mean VMR are 61 (IAGOS) and 31 (ERA). Is the value for ERA correct?

P9 l15: where do you compare 'the seasonal cycle'?

[Figure]

P9 l16: there is larger variability in the in-situ data. Why do you say 'smaller variability'?

P12 l6: correct 'profil' with 'profile'.

P14 l13: correct 'similiar' with 'similar'.

Reference

Eleftheratos, K., Zerefos, C. S., Zanis, P., Balis, D. S., Tselioudis, G., Gierens, K., and Sausen, R.: A study on natural and manmade global interannual fluctuations of cirrus cloud cover for the period 1984–2004, Atmos. Chem. Phys., 7, 2631-2642, 2007.
* * *

---

## Author Comment (AC1) · 24 Oct 2019

**Reply to Reviewer**

We thank both referees for their insightful reviews and helpful and constructive comments.

We agree that more scientific significance should be added to the manuscript. Therefore, we extended the chapter about pathlengths of ice supersaturated regions to include new findings and characteristics of pathlengths and distances between ISSRs.

The structure of the manuscript was therefore changed. Now, Section 3 focuses on the validation of basics variables of ERA with IAGOS measurements. Section 4 deals with the horizontal scales of ice supersaturated regions. We compare the statistics of pathlengths not only to ERA but also to higher resolution measurement data. The seasonal cycle and height dependence are also investigated.

We also made clear, that this manuscript is part of a joint investigation of water vapour in the UTLS region. A companion study by Petzold et al. (2019, in review) is focussing on the physical interpretation of the water vapour distribution in the UTLS region. Beside the investigation of the seasonal cycle of RHi and ISSRs, they study the correlation between NAO and ISSR occurrence and trend analyses.

Also, some minor errors in the analysis tools were found and corrected. First, to be consistent with other studies, an ISSR is now omitted from the statistics, as soon as the pressure level of the airplane is changed by more than 50 Pa. Additionally, a typo in the geographical mask was found and corrected. Instead of -65.E to **5.0E** the algorithm used -65.0E to **0.5E**. Also, the time frame of the used data is shortened to 2000 to 2009 due to a very low number of flights in 2010. In combination, the total number of ISSRs is reduced compared to the first version. Also the mean and median values slightly changed. However, the conclusions are not affected by this.

In the following, we answer to the comments point by point. *Questions and remarks of the reviewers are marked in orange,* reply of the authors are marked in black and *changes to the manuscript are marked in blue.*

At the end of this document the final version of the revised manuscript can be found. Here, red marks deleted parts of the previous version and blue indicates new parts.

**Reviewer #1**

**Reviewer:** *I would use a more applicable title such as: Validation of ECMWF ERA-Interim Reanalysis with IAGOS in-situ water vapour measurements in the UTLS region.*

**Authors:** We agree to use a more applicable title. Since we extended also the sections including pathlengths of ice supersaturated regions, we choose following title: *Ice supersaturated regions: properties and validation of ERA-Interim Reanalysis with IAGOS in-situ water vapour measurements*

**Reviewer:** *How may data points does your comparison involve (and how many in each level)? Instead of the PDFs in Fig. 1 you could plot the total number of measurements per gridbox.*

**Authors:** We replaced Fig. 1 and are now using a map with the total number of measurements. The area of interest is divided into 35 x 35 gridboxes for this visualization. The previous version of Fig. 1 is also available in the publication by Petzold et al. (2019, in review)

[Figure]

*Figure 1: Number of IAGOS measurements per gridbox (2° in longitudinal and 0.57° in latitudinal direction) during January 1st, 2000 to December 31st, 2009.*

We also added the information of the number of measurements in each level in Table 1:

| Region | Shortname | $p_{ap} - p_{tph}$ [hPa] | number of measurements |
|---|---|---|---|
| | LS3 | -90 | 3 203 483 |
| Lowermost stratosphere | LS2 | -60 | 4 237 245 |
| | LS 1 | -30 | 5 268 138 |
| Tropopause layer | TL | 0 | 5 643 057 |
| | UT1 | +30 | 4 649 883 |
| Uppermost troposphere | UT2 | +60 | 2 647 935 |
| | UT3 | +90 | 909 120 |

**Table 1.** The data set is distributed into three main layers: the upper troposphere, tropopause layer, and lowermost stratosphere. The outer layers are additionally subdivided into three sublayers. The distribution criterion is the pressure difference between aircraft pressure $p_{ac}$ and the tropopause pressure $p_{tph}$ with the range of $\pm 15$hPa. Additionally, for every flight layer the number of IAGOS measurements between 2000 and 2009 are presented.

**Reviewer:** *I don't see much added value of showing PDFs and box plots of the same data especially because they are not interpreted in detail each. I would prefer the box plots in the paper and to just mention the shape of the PDFs where required. Showing the two data sets in one box plot side by side would make the comparison easier*

**Authors:** We omit Fig. 2 and Fig. 4 (PDFs of temperature and water vapour, respectively) in order to streamline the validation. Additionally, we redesigned Fig. 3, Fig. 5. and Fig. 6. to have the box plots side by side

[Figure]

*Figure 2: Vertical distribution of temperature for IAGOS (left) and ERA (right). Improved design.*

[Figure]

*Figure 3: Vertical distribution of water vapour volume mixing ratio (VMR) for IAGOS (left) and ERA (right). Improved design.*

[Figure]

*Figure 4: Vertical distribution of relative humidity over ice for IAGOS (left) and ERA (right). Improved design.*

**Reviewer:** *Fig. 2: Could be the non-Gaussian distribution of temperature values in UT3 be a result of the used tropopause definition?*
and
**Reviewer:** *The wave-like structure in Fig. 2 (mentioned on p. 7 l. 3) would be worth investigating to increase the scientific statement of the paper, especially because it is not only visible in UT3, but continues to UT2 and UT1. You could analyse the data of the discrete steps or waves, separately and explore their properties (see scientific suggestions above).*

**Authors:** Here we present also the results of the temperature probability function in dependence of the tropopause definition.

[Figure]

*Figure 5: Probability density function of temperature for different heights (coloured) for IAGOS (left) and ERA (right) for the __dynamical__ tropopause definition.*

[Figure]

*Figure 6: Probability density function of temperature for different heights (coloured) for IAGOS (left) and ERA (right) for the __thermal__ tropopause definition.*

We focus in the revised version of the manuscript more on the pathlengths of ice supersaturated regions. Therefore, we omit the display of probability functions (PDF) of temperature and mixing ratio (Fig.2 and 4, respectively).

Nevertheless, we want to give a short reply on your question about the influence of the tropopause definition on the behaviour of the PDFs. Figure 5 shows the vertical distribution of temperature for IAGOS and ERA using the dynamical tropopause definition. Here, the lowest layer UT3 (pink line) shows a strong non-Gaussian distribution. In Fig. 6 the thermal tropopause definition after WMO (1957) is used. Here, the lowest level shows a clearly smoother behaviour. However, this effect is stronger for the IAGOS measurements than for ERA.

The difference in the vertical structure due to the tropopause definition is also visible for the stratospheric layers which are shifted to significantly warmer temperatures (green, blue and brown lines). This is because the thermal tropopause is usually higher than the tropopause using the dynamical definition.

However, we do not unravel this feature in more detail in this manuscript, since we focus here on water vapour and ice supersaturation. But we keep this feature in mind for future studies. We also extended the manuscript regarding the pathlengths of ISSR to increase the scientific significance of this investigation.

**Reviewer:** *Dyroff et al. 2015 found a moist bias in ECMWF analyses and forecasts in the lower stratosphere. Please discuss why this bias is not apparent in your data.*

**Authors:** The IAGOS Capacitive Hygrometer (ICH) uses a capacitive sensor (HUMICAP® of type H, Vaisala, Finland). CARIBIC measure $H_2O$ with a combination of a frost point hygrometer and a photo-acoustic hygrometer, which can detect volume mixing ratios in the stratosphere below 5-10 ppmv with a better precision and uncertainty than ICH. Therefore, the uncertainties in volume mixing ratios of $H_2O$ are getting relatively large and may explain the more wet stratospheric values compared to CARIBIC.

Based on pre-and post-flight calibrations, the ICH reports RH data with an uncertainty of 4% RH in the middle troposphere and 6% RH at the tropopause (Smit et al., 2014). Applying the 2-sigma criterion (95% confidence level), the ICH limit of detection (LOD) is $RH_{ice,LOD}$ = 12% which transfers in a minimum detectable $H_2O$ VMR of approx. 10 ppmv at typical mid-latitude upper troposphere conditions (T = 218K, p = 250 hPa).

This is in agreement with in-flight intercomparisons against high-precision water vapour instruments in dedicated research aircraft studies (Helten et al., 1999; Neis et al., 2015a; Neis et al., 2015b). More detailed information on the instrumentation of IAGOS is also available in the accompanying manuscript by Petzold et al. (2019, in review)

We include following text to the manuscript:

*Dyroff et al. (2015) reported a moist bias comparing CARIBIC measurements with ECMWF analyses and forecasts. Since ERA-Interim is based on ECMWF analyses one would expect also a moist bias in the lower stratosphere. In contrast to the capacitive sensor used for IAGOS, CARIBIC uses a combination of a frost point hygrometer and a photo-acoustic hygrometer, which shows a better precision and uncertainty for very low water vapour volume mixing ratios. Therefore, the uncertainties in volume mixing ratios of H2O are large in the lower stratosphere and may explain the more wet stratospheric values compared to CARIBIC.*

Finally, the distribution of water volume mixing ratio for IAGOS and ERA is seen in the following figure, where a slight shift to larger values is visible in the two uppermost layers (LS2 and LS3) for ERA.

[Figure]

*Figure 7: Histogram of water vapour mixing ratio for all levels (from LS3 at the top to UT3 at the bottom) for IAGOS (blue) and ERA (red). For the layers LS2 and LS3 the maximum of ERA shifted to higher mixing ratios.*

**Reviewer:** *P. 11 l. 7: The sentence is a bit weird: Do you call ERA-Interim a climate model or do you draw from your results that climate models must show an underestimation of ISSRs as well? Please reconsider/reformulate*

**Authors:** Yes, this sentence is not well written. We wanted to express a general implication of the underestimated abundance of ISSRs. We reformulated it to:

*"Since ISSRs in the lower stratosphere are an important factor for the lifetime of contrail cirrus, a good model representation of the abundance of ice supersaturation in this region is important for an adequate description of the Earth's radiative budget."*

**Reviewer:** *Do the negative RHi for IAGOS in the LS in Fig. 6 make sense?*

**Authors:** Before the implementation into the aircraft each sensor is calibrated. For a new calibrated sensor in laboratory conditions the offset is zero. This offset is changing over time by deposition of dirt on the sensor, which leads to a mostly negative offset. Although this drift is corrected, due to the measuring inaccuracy values down to -5% for the relative humidity with respect to water are possible. This converts to -15% for $RH_i$ at the most.

More information on data processing can be found in Petzold et al. (2019, in review).

**Reviewer:** *How does your statistics of temperature and mixing ratio (Fig. 3, 5) appear inside (and outside) of the ISSRs? Is there a way to relate the underestimation of RHi>100% in ERA to either of these variables?*

**Authors:** We add here the vertical profiles for temperature (Fig. 8) and water volume mixing ratio (Fig. 9) for cases, were both IAGOS and ERA show ice supersaturation at the same time.

For the highest level LS3 no overlapping ISSRs occur. In layer LS2 ERA shows colder temperatures, but Figure 9 also reveals dryer conditions for ERA compared to IAGOS. Both features (colder and dryer) are compensating each other.

In the layers beneath no significant deviations between IAGOS and ERA are visible. The reason for this might be that only data points were used, where both IAGOS and ERA show supersaturation. This is most likely the case within an ice supersaturated region, were both data sets agree well.

Apparently, the differences appear, where both datasets are not overlapping, for instance at the edge of ISSRs. Figure 8 of the revised manuscript shows the exemplary flight from Atlanta (USA) to Frankfurt (Germany), where ERA shows a good agreement with IAGOS as far as the large-scale variability is concerned. However, the positions of the ISSR in that case are shifted in ERA, especially for the second ISSR around a flown distance of 6000 km. Therefore, the agreement between IAGOS and ERA is quite good when both data sets have an ISSR at the same position.

[Figure]

*Figure 8: Vertical profile of temperature for IAGOS (blue) and ERA (red) for cases, where both IAGOS and ERA show ice supersaturation.*

[Figure]

*Figure 9: Vertical profile of water vapour volume mixing ratio for IAGOS (blue) and ERA (red) for cases, where both IAGOS and ERA show ice supersaturation.*

**Reviewer:** *P. 12 l. 6ff and Fig. 8: Does your ISSR fraction mean the (relative) number of ISSRs among all data points? Please clarify.*

**Authors:** The ISSR fraction is defined as the number of data points with RHi ≥ 100% within a layer divided by the number of all data points in that layer. We added following sentence:

*"The ISSR fraction in this study is defined as the number of data points within a layer with RHi ≥ 100% divided by the total number of data points in that layer above the defined North Atlantic region."*

**Reviewer:** *subsection 3.4.1: What are the flight distances of your examples? It would prefer a length axis instead of or in addition to time in Fig. 9*

**Authors:** We have changed this part and present only one exemplary flight. We use the flown distance as x-axis.

**Reviewer:** *subsection 3.4.1: To add some scientific content it would be very intersting to discuss the weather situation behind the flights, e.g. with a satellite image.*

**Authors:** We have added a discussion of the weather situation for the specific flight.

**Reviewer:** *As far as I understood the first paragraph of 3.4.2 refers to the comparison of IAGOS (1km) and ERA in Fig. 10? If so, it would help to draw the reader's attention to the respective lines in the figures e.g. (see black and red lines in Fig. 10).*

**Authors:** We have added this to the text.

**Reviewer:** *P. 14 l. 8: "Already with a resolution of 1km the cumulative distribution exhibits a different character." Different to what? Or do you mean "10km" instead of "1km"?*

**Authors:** Yes, this was a typo. We reformulated this passage to:

*"Already with a resolution of 2 km the number of small ISSRs is decreasing strongly. The median value increases to 7 km, compared to the original resolution. Decreasing the spatial resolution further leads to a decrease in the number of found ISSRs and an increase of the mean and median ISSR pathlength. At a resolution of 10 km the cumulative distribution exhibits a different character compared to the original data."*

**Reviewer:** *P. 15 l. 1: What exactly do you mean by: "...the behaviour of the ERA data set changes for pathlengths of 100km..."? Please revise*

**Authors:** We wanted to point out that the course of the cumulative distribution for ERA is different for pathlengths smaller than 100 km compared to the 100 km running mean of IAGOS. We reformulated this part to:

*"When comparing the results of the 100km running mean of IAGOS with the ERA data set it is also noteworthy that the behaviour for ERA is clearly different for pathlengths smaller than 100 km. ERA shows a significantly higher probability for very small ISSRs (< 10 km) than the running mean with 100 km. However, the increase in probability to pathlengths of up to 100 km is flatter in ERA."*

**Reviewer:** *P. 16 l. 8-11: This is part of the motivation and should appear much earlier, in the introduction*

**Authors:** Yes, we moved this part to the introduction.

*Additional comments:*
*1. P. 1 l. 23: Remove the "s" from clouds*
We corrected this.

*2. P. 2 l. 25: hyydroxyl?*
We corrected this.

*3. P. 9 l. 14: "that IAGOS data set shows": add "the" before IAGOS or remove "set"*
This part was rewritten.

*4. P. 11 l. 11: RHi>1: Do you mean RHi>100%?*
We mean 100%. We corrected this.

*5. P. 12 l. 9: Remove double "the"*
We corrected this.

*6. P 12 l. 14: Correct "reanaylsis"*
This part was rewritten.

*7. P. 14 l. 10: Delete one "s" in "median values increases"*
This part was rewritten.

*8. P. 15 l. 5. Lower case "s" in ISSRS*
We corrected this.

*9. P. 15 l. 26: I would change "Moving up to the stratospheric layers, as expected, the RHi values are" to "...layers the RHi values, as expected, are*
We corrected this.

**Reply to Reviewer #2**

**Reviewer:** *The study shows the good ability of IAGOS measurements to capture small scale ISSRs (smaller than 100 km) and at the same time gives credit to the ERA-Interim model to depict large scale ISSRs (larger than 100km). These are important findings which merit publication as far as our knowledge on the detection of ISSRs from different datasets. However, presenting only pdfs and boxplots in a comparison study is not sufficient to justify publication in a journal as such ACP. The authors should make deeper comparisons with their ISS data. For instance:*

*1) They could perform simple time series analysis for their region (40-60N, 5-65W) and plot the monthly time series of the two ISS datasets from 2000 to 2010, compare the seasonal cycles, and then correlate the two time series after removing the seasonal variability.*

**Authors:** We thank the reviewer for the constructive suggestion.

We decided to increase the scientific value by investigating the pathlengths of ice supersaturated regions. Additionally, we added a comparison of the timeseries of the fraction of ISSR for the upper troposphere, tropopause region and lower stratosphere. Since in the accompanying study by Petzold et al. (2019, in review) (de-seasonalized) time series of ISSRs are shown, we focus on the comparison between IAGOS and ERA. We show that ERA can reproduce the ISSR fraction if a lower threshold than $RH_i$=100% is used. In the lower stratosphere the best agreement between IAGOS and ERA is found, when ice supersaturation in ERA is defined as $RH_i \geq 85\%$. In the tropopause this threshold increases to $RH_i \geq 90\%$ and in the upper troposphere to $RH_i \geq 95\%$ (dashed red lines in Fig. 10).

[Figure]

*Figure 10: Time series of monthly fraction of ice supersaturated regions for IAGOS (blue) and ERA (red) for the lower stratosphere (LS3 to LS1), the tropopause layer (TL), and the upper troposphere (UT1-UT3). The dashed red line represents the ERA data set with the fraction of $RH_i \geq 85\%$ for the lower stratosphere, $RH_i \geq 90\%$ for the tropopause layer and $RH_i \geq 95\%$ for the upper troposphere. Note the different scale for the lower stratosphere.*

**Reviewer:** *2) It has been shown that cirrus cloud variability is significantly affected by the North Atlantic Oscillation during winter (Eleftheratos et al., 2007). The authors could test if a correlation between the deseasonalized ISS data and the NAO index exists.*

**Authors:** The accompanying publication by Petzold et al. (2019, in review) investigates the correlation between the NAO-Index an the IAGOS data. The authors conclude that there is a statistically significant correlation for the North Atlantic and Europe.

We have made it clearer in the manuscript that our study is part of a joint investigation of water vapour in the upper troposphere to the lower stratosphere and added the following part to the introduction:

*"This study is part of a joint investigation of water vapour in the upper troposphere to the lower stratosphere. A companion20study by Petzold et al (2019) will focus on the physical interpretation of the water vapour distribution in the UTLS region. There, a detailed investigation of the seasonal cycle of $RH_i$ and ISSRs, the physio-chemical signature of ISSR, the ISSR fraction and cirrus cloud occurrence is presented. Additionally, they also present a trend analysis."*

*Technical corrections*
*P3 l16: correct 'asses' to 'assess'.*
We corrected this.

*P3 l26: correct 'continous' to 'continuous', correct 'greenhous' to 'greenhouse'.*

We corrected this.

*P4 l2: you say '40N-60N, 5-65W' but on p5 l16 you write 'from 40o to 60o North and -65o to 5o East'. Please write the correct coordinates for longitude.*

We corrected this and write now (40N to 60N, -65E to 5E)

*P4 l15: what is the '4s resolution'?*

We mean the temporal resolution of the data. We clarified this in the text.

*P8 table 3: for the case of TL, the mean VMR are 61 (IAGOS) and 31 (ERA). Is the value for ERA correct?*

This is a typo. The correct value is 58 ppmv for ERA.

*P9 l15: where do you compare 'the seasonal cycle'?*

We deleted this fragment of an earlier manuscript version.

*P9 l16: there is larger variability in the in-situ data. Why do you say 'smaller variability'?*

We reformulated it to: *"In summary, the reanalysis data is in good agreement with the vertical distribution of the IAGOS data. However, IAGOS shows a larger variability and stronger extreme values."*

*P12 l6: correct 'profil' with 'profile'.*

This part was rewritten.

*P14 l13: correct 'similiar' with 'similar'.*

This part was rewritten.

References:

WMO (World Meteorological Organization): Meteorology – A three-dimensional science: Second Session of the Commission for Aerology, WMO Bull. IV(4), WMO, Geneva, 134–138, 1957.

[revised manuscript text omitted]

---

## Referee Report (RR1)

Review of the 2nd submitted version of:

**"Ice supersaturated regions: properties and validation of ERA-Interim Reanalysis with IAGOS in-situ water vapour measurements"**

With the first revision the authors improved the scientific content by adding some further comparisons of the ISSR pathlengths. However, the new subsections are not very carefully written concerning the discussion of the figures and the language. I would recommend to accept the manuscript after respecting the following comments that mainly address the new sections:

Major:

1. The time series in Fig. 7 does not have an additional value without describing the different years, seasons or months. According to your text it would be sufficient to additionally plot lines for RHi=85, 90 and 95% into Fig. 6. When you want to keep Fig. 7 you should discuss the time series in more detail.

2. Related to Fig. 10 only the pathlengths gained from the measurements are discussed. You must decide if this section should include the validation of ERA or not. If not, the lines for ERA in Fig. 10 should be removed. Otherwise, ERA must be discussed as well.

3. Do you really need Fig. 12 for your argumentation? So far, table 7 provide the numbers you use in the text. Furthermore, this Fig. would only provide the median and not the mean you mostly argue with. If you want to keep the figure you should carefully discuss all features they show, including the comparison with ERA, which is again not even mentioned here (see previous comment).

Minor:

1. Figs. 2, 3 and 4 should still be improved by directly juxtaposing the box and whiskers of IAGOS and ERA, which means one shrink the two x-axes to one. Having the boxes next to each other would allow a direct and quantitative comparison of the data sets.

2. P. 10 l. 5: The contrail cirrus arrives out of the blue here. You should at least provide a reference, or even introduce the importance of ISSRs for contrails in the Intro.

3. Figs. 4 and 5: Could one hypothesize that ice clouds in ERA are produced too "early" which decreases supersaturation? You might think about a more constructive comment than "…points to a missing process…." on p. 11 line 1.

4. P. 14 l. 4/5: You should introduce the warm conveyor belt and add a reference, at least similar to the dry intrusion a few lines after.

5. Fig. 8 upper panel: It would be nice to cut the satellite image at the left and right side to focus on the region of interest.

6. Fig. 8 lower panel: Interestingly, the supersaturation is even higher in ERA compared to IAGOS for the second ISSR in this particular case! It would be very helpful to see the height of the 2pvu-line to visually assign the region where the ISSRs occur (if UT, TP or LS).

7. The caption of Fig. 9 appears preliminary.
8. The title of subsection 4.2 seems a bit weird. What do you mean by "ISSR extensions"? Please explain!

9. Fig. 11: Why don't you show a box and whisker for each season, with an additional marker where the mean is located? This would add some information about the distribution of the values, which would also be nice to be addressed in the text.

10. P. 21 l. 10 ff: I don't understand this section: "A shorter mean or median pathlength implies that ISSRs are closer to each other (D14). Therefore, in winter the conditions favour ISSRs with small distances." Wasn't the summer in your data the season with the shortest ISSR pathlengths? Something seems to be wrong, either in the citation or in the season you mentioned. Also in combination with the last sentence of this paragraph : "Therefore, not only the distance between ISSR pathlengths is shortest, but also the pathlength is largest". Wouldn't this be contradictory with the sentence written before? Please rephrase or clarify in the text.

11. The titles of chapter 4 appear inhomogeneous, or rather 4.1 too technical. My suggestions would be: 4.1. Systematic (or statistic) investigation of ISSR pathlengths; 4.2 Seasonal cycle of ISSR pathlengths and 4.3. Seasonal cycle of the distances between ISSRs.

12. Are there already ideas for the reasons of the pathlengths in the different seasons and heights? Would be great to have at least some speculations to enhance the connection of your results to the meteorology.

13. Your final phrase is not a nice end. It directs the reader's attention to ERA5 while the value of your study takes the back seat.

Additional:

1. P. 2 l. 5: "The effect of cirrus clouds are ….": Replace are with is.

2. P. 3 l. 3: "…description of processes …are…." Replace are with is.

3. P. 3 l. 26: ".., study…focuses" instead of "focus"

4. P. 10 l. 10: Write : "… with cumulative probability in Figure 5." and delete the sentence thereafter.

5. P.12 l. 10: Verb missing : "One reason might -be- the data assimilation…"

6. P. 13 l. 4 and 5: Rephrase the two sentences, they are not really clever….

7. P. 14 l. 13: remove double "in"

8. P. 14 l. 34: "…,big difference.s While….."

9. P. 16 l. 5: Add "s" to "value"

10. P. 16 l. 8: "an original" instead of "a original"

11. P. 17 l. 7: remove double "character"

12. P. 17 l. 18: "The previous results discussed…." should mean "… we discussed…"? Results can not discuss….

13. P. 17 l. 18: The "uniqueness" IAGOS data is already mentioned on P. 14 l. 19. Please delete it here or there.

14. P. 17 l. 21: an "o" is missing in "troposphere"

15. P. 17 l. 23: "…fraction….are present…": is instead of are

16. P. 17 l. 28: "in the summer": delete "the"

17. P. 17 l. 31: double ".."

18. P. 18 l. 11: "…in summer an the highest…" should be "and"

19: P. 21 l. 6: add "s" to layer

20. P.21 l. 10: replace the citation by the abbreviation "D14" you introduced earlier

21. P. 21 l. 11: add "s" to suggest

22. P. 21 l. 22: missing "i" in within

23. P. 23 l. 18: capital "S" for plural in ISSRS

24. P. 23 l. 32: close parenthesis after "..31 km"

---

## Author Response (AR2)

**Reply to Reviewer**

We thank the reviewer again for the remarks and suggestions. We appreciate her or his careful review.

In the following, we answer the comments point by point. *Questions and remarks of the reviewers are marked in orange,* reply of the authors are marked in black and *changes to the manuscript are marked in blue.*

At the end of this document the final version of the revised manuscript can be found. Here, red marks deleted parts of the previous version and blue indicates new parts.

**Major comments**

**Reviewer:** *1. The time series in Fig. 7 does not have an additional value without describing the different years, seasons or months. According to your text it would be sufficient to additionally plot lines for RHi=85, 90 and 95% into Fig. 6. When you want to keep Fig. 7 you should discuss the time series in more detail.*

**Authors:** We included the lines for RHi=85,90 and 95% to Fig. 6 as well. We also added more discussion of Fig. 7 and referred also to the companion study by Petzold et al. (2019). Therefore, we would like to keep Fig. 7 as well.

[Figure]

**Figure 6.** Vertical profile of the fraction of ice supersaturated regions for IAGOS (blue) and ERA (red). Different definitions of *ice supersaturation* in ERA are shown in light red. The dotted line represents the fraction of $RH_{i_{ERA}} \geq 85\%$, the dashed-dotted line depicts the fraction of $RH_{i_{ERA}} \geq 90\%$ and the dashed lines stands for the $RH_{i_{ERA}} \geq 95\%$.

*Figure 1: New Figure 6 including the lines for RHi=85,90 and 95%.*

5  Comparing the time series with Fig. 6, it is again visible that ERA is underestimating the fraction of ISSR compared to the measurements from IAGOS.  a different $RH_{i_{ERA}}$ threshold for the ERA data set leads to the best agreement.  Using a different definition of *ice supersaturation* by reducing the threshold in ERA (85, 90 and 95 %) improves the agreement between ERA and IAGOS in each layer. However,

10  the different threshold can also lead to overestimation of the ISSR fraction. While in the lower stratosphere , a threshold of $RH_{i_{ERA}} \geq 85\%$
* * *
 shows a overall good agreement, in 2005 this threshold would lead to an overestimation of the ISSR fraction of up to 300 % (December 2005). It is also noteworthy that, although Fig. 6 shows an almost perfect agreement between IAGOS and the modified threshold of 95 % in the upper troposphere

5   large deviations between IAGOS and ERA can be found as well, for example in 2004.

As a side note, the reader is referred to the companion study by Petzold et al. (2019), where also a trend analysis using

10  the IAGOS data was conducted, which led to the conclusion that no significant trends in ISSR occurrence can be observed. However, they do find a correlation between the North Atlantic oscillation (NAO) and ISSR occurrence.

*Figure 2: Screenshot of the modified text regarding FIg. 7 in the manuscript.*

**Reviewer:** *2. Related to Fig. 10 only the pathlengths gained from the measurements are discussed. You must decide if this section should include the validation of ERA or not. If not, the lines for ERA in Fig. 10 should be removed. Otherwise, ERA must be discussed as well.*

**Authors:** You are right. We added the discussion of ERA as well in the text.

The results for ERA (dotted lines) in spring, and also for the other seasons, are clearly different. As mentioned above, short ISSRs are underestimated by ERA due to the spatial resolution, which is visible for all seasons. The different resolution leads also to other relations of the pathlengths between different height levels. For instance, the highest fraction of ISSR in spring with a pathlength between 10 km and 100 km is found in the lower stratosphere. This behaviour is vice versa to the IAGOS data set, where the upper troposphere showed the highest fraction. The same observation can be made in winter as well. Not surprisingly, ERA misses also the high fraction of short ISSRs in the stratosphere during the summer season.

In general, the mean pathlength of an ISSR in the IAGOS data set is significantly larger in the stratosphere compared to the upper troposphere and tropopause layer, except for the summer season, which is also shown in Fig. 11. Comparing the mean pathlengths for every season in the IAGOS data makes clear that the strongest seasonal cycle is present in the lower

*Figure 3:Screenshot of the modified text regarding Fig. 10 in the manuscript.*

**Reviewer:** *3. Do you really need Fig. 12 for your argumentation? So far, table 7 provide the numbers you use in the text. Furthermore, this Fig. would only provide the median and not the mean you mostly argue with. If you want to keep the figure you should carefully discuss all features they show, including the comparison with ERA, which is again not even mentioned here (see previous comment).*

**Authors:** Again, you are right. We added the following discussion of ERA in the text and therefore also keep Fig. 12 in the manuscript.

The results for ERA (dotted lines in Fig. 12) show again a clearly different picture. The comparison of the height dependence

30     between IAGOS and ERA reveals a reversed and even enhanced dependence as it is shown by the distance of the vertical layers  within the ERA data set. For instance in winter, the reversed dependence with height is most prominent. Between 1 km and 100 km the lower stratosphere shows the highest fraction of distances, followed by the tropopause layer and the upper troposphere. In ERA, this dependence is vice versa, with the highest fraction in the upper troposphere and the lowest fraction in the lower stratosphere.

*Figure 4:Screenshot of the modified text regarding Fig. 12 in the manuscript.*

**Minor comments**

**Reviewer:** *1.Figs. 2, 3 and 4 should still be improved by directly juxtaposing the box and whiskers of IAGOS and ERA, which means one shrink the two x-axes to one. Having the boxes next to each other would allow a direct and quantitative comparison of the data sets.*

**Authors:** We changed the design of the Figs. 2,3 and 4 and plot the boxes next to each other. Please find below the new version of the Figs.

[Figure]

Figure 2. Vertical profile of temperature [K] for IAGOS (left blue) and ERA (right red).

[Figure]

Figure 3. Vertical profile of $H_2O$ volume mixing ratio [ppmv] for IAGOS (left blue) and ERA (right red).

[Figure]

Figure 4. Vertical profile of relative humidity with respect to ice (RHi) [%] for IAGOS (left blue) and ERA (right red).

**Reviewer:** *2. P. 10 l. 5: The contrail cirrus arrives out of the blue here. You should at least provide a reference, or even introduce the importance of ISSRs for contrails in the Intro.*

**Authors:** We added the reference of Kärcher (2018) and a short statement in the introduction.

**1 Introduction**

Water vapour is the most important greenhouse gas in the atmosphere and therefore plays a major role in the Earth's radiative balance (Myhre et al., 2013). Especially in condensed form water is also of large significance for the planetary radiation. Clouds can reflect incoming solar radiation, while absorbing and reemitting longwave radiation from the earth. Particularly

5    the effect of cirrus clouds  is still challenging. Whether a cirrus cloud has a net warming or cooling effect on the Earth's atmosphere depends strongly on altitude, available humidity and microphyiscal properties like number, size and type of ice nuclei (IN). Even the same exact cirrus cloud can change the sign of its net forcing depending on the time of day (Joos et al., 2014). Beside natural cirrus clouds also the aircraft-induced contrail cirrus clouds play an important role for the radiative budget (Kärcher, 2018).

*Figure 5: Screenshot of the added sentence about the importance of contrail cirrus.*

Kärcher, B.: Formation and radiative forcing of contrail cirrus, Nature Communications, 9, https://doi.org/10.1038/s41467-018-04068-0,http://www.nature.com/articles/s41467-018-04068-0, 2018.

**Reviewer:** *3. Figs. 4 and 5: Could one hypothesize that ice clouds in ERA are produced too "early" which decreases supersaturation? You might think about a more constructive comment than "…points to a missing process…." on p. 11 line 1.*

**Authors:** We reformulated this part to:

soon as they reach ice supersaturation. As mentioned before, the IFS-model allows the existence of ice supersaturation, but only in cloud free conditions. As soon as ice clouds are present in the models grid cell the supersaturation is adjusted to $RH_i = 100\%$. Unfortunately, the IAGOS data set of the investigated time frame cannot distinguish between cloudy and non-cloudy areas. [1] Nevertheless, also in ice clouds ice supersaturation is present (Krämer et al., 2016). Therefore, the behaviour of

5 the cumulative distributions for ERA, especially in the layers from UT3 to TL,  might be due to an untimely formation of ice clouds in the underlying IFS model, which adjusts the ice supersaturation too early.

Another way to compare the representation of the water vapour is the fraction of ISSRs. Figure 6 presents the vertical profile of the ISSR fraction. The ISSR fraction in this study is defined as the number of data points within a layer with $RH_i \geq 100\%$

10 divided by the total number of data points in that layer above the defined North Atlantic region. It is clearly visible that the measurements by IAGOS show a higher fraction of ISSR. Only for the two uppermost layers the fraction of both data sets are of comparable magnitude. Here, the very dry conditions produce only in very few cases supersaturation. The largest difference
* * *
[1]Nowadays, the IAGOS setup includes an optical sensor for registration of clouds on the flight path.

*Figure 6: : Screenshot of the reformulated part of the manuscript..*

**Reviewer:** *4. P. 14 l. 4/5: You should introduce the warm conveyor belt and add a reference, at least similar to the dry intrusion a few lines after.*

**Authors:** We added the reference of Spichtinger et al., (2005) regarding the warm conveyor belt and added the following part to the manuscript:

25 UTC, while the satellite image is taken at 12 UTC, therefore a shift between measurements and image has to be kept in mind. A low pressure system, located between the British Isles and Iceland is dominating the weather over the North Atlantic. The warm conveyor belt (WCB), an ascending airflow from the boundary layer to the upper tropopause (Spichtinger et al., 2005), with its cloud band is clearly visible reaching from Ireland to the south-west. Between a flown distance of 5000 to 6000 km an

*Figure 7: Screenshot of added text regarding the warm conveyor belt.*

**Reviewer:** *5. Fig. 8 upper panel: It would be nice to cut the satellite image at the left and right side to focus on the region of interest.*

*And*

*6. Fig. 8 lower panel: Interestingly, the supersaturation is even higher in ERA compared to IAGOS for the second ISSR in this particular case! It would be very helpful to see the height of the 2pvu-line to visually assign the region where the ISSRs occur (if UT, TP or LS).*

**Authors:** We cropped the image and added the 2pvu-line into the figure:

[Figure]

**Figure 8.** Example of typical flight from Atlanta (USA) to Frankfurt (Germany) on March 7th, 2009. The upper part shows the synoptic situation over the North Atlantic on that day including the flight path. For better orientation red dots are placed every 1000 km. In the lower part RH$_i$ is shown over the distance of the flight for IAGOS (blue) and ERA (red). Shaded areas denote ice supersaturation. The pressure level of the aircraft is shown in black and the height in hPa of the dynamical tropopause (2PVU) is shown in green. Satellite image: copyright 2009 EUMETSAT.

*Figure 8: Reworked Figure 8 of the manuscript.*

**Reviewer:** *7. The caption of Fig. 9 appears preliminary.*

**Authors:** We corrected this.

**Reviewer:** *8. The title of subsection 4.2 seems a bit weird. What do you mean by "ISSR extensions"? Please explain!*

**Authors:** Please see remark 11. We reformulated the titles.

**Reviewer:** *9. Fig. 11: Why don't you show a box and whisker for each season, with an additional marker where the mean is located? This would add some information about the distribution of the values, which would also be nice to be addressed in the text*

**Authors:** We reworked Fig. 11 and added reformulated the text.

values are significantly larger compared to IAGOS. More importantly, the seasonal cycle of ERA is shifted, especially for the
10  tropopause and lower stratosphere. The maximum mean pathlength in the tropopause can be found in summer for IAGOS, where in the ERA data the minimum pathlength is found. The same holds true for the lower stratosphere, where the minimum for IAGOS is also found in summer, where ERA exhibits the maximum. Additionally, the  relative amplitude of the seasonal cycle is smaller compared to IAGOS, most prominent in the lower stratosphere. Here, also the spread of the
15  data, indicated by the size of the boxes, shows a large fluctuation in the IAGOS data set, which is not caputred by ERA.

*Figure 9: Screenshot of the reformulated text regarding Fig. 11.*

[Figure]

**Figure 11.**  Box-and-Whisker plot of the seasonal cycle of mean and median  pathlengths of ISSR for different atmospheric layers in the IAGOS data set. Solid blue line shows the lower stratosphere (LS), solid black line the tropopause layer (TL), and solid red line upper troposphere (UT) for the IAGOS data set. Light colors denote the results for the ERA data set. Dashed lines denote the median pathlength, respectively. Note the break in the vertical axis.

*Figure 10: Reworked Fig. 11.*

**Reviewer:** *10. P. 21 l. 10 ff: I don't understand this section: "A shorter mean or median pathlength implies that ISSRs are closer to each other (D14). Therefore, in winter the conditions favour ISSRs with small distances." Wasn't the summer in your data the season with the shortest ISSR pathlengths?*

*Something seems to be wrong, either in the citation or in the season you mentioned. Also in combination with the last sentence of this paragraph: "Therefore, not only the distance between ISSR pathlengths is shortest, but also the pathlength is largest". Wouldn't this be contradictory with the sentence written before? Please rephrase or clarify in the text.*

**Authors:** The confusion was caused by a typo. We meant "distance" and not "pathlength". Now, this part makes sense:

*"A shorter mean or median  **distance** implies, that ISSRs are closer to each other"*

**Reviewer:** *11. The titles of chapter 4 appear inhomogeneous, or rather 4.1 too technical. My suggestions would be: 4.1. Systematic (or statistic) investigation of ISSR pathlengths; 4.2 Seasonal cycle of ISSR pathlengths and 4.3. Seasonal cycle of the distances between ISSRs*

**Authors:** We followed your suggestions and renamed the sections to:

*4 Horizontal scales of ice supersaturated regions in IAGOS and ERA-Interim*

*4.1 Statistic investigations of ISSR pathlengths.*

*4.2. Seasonal cycle and height dependence of ISSR pathlengths*

*4.3 Seasonal cycle and height dependence of the distances between ISSRs*

**Reviewer:** *12. Are there already ideas for the reasons of the pathlengths in the different seasons and heights? Would be great to have at least some speculations to enhance the connection of your results to the meteorology.*

**Authors:** We added following speculation (in blue):

stratosphere with the lowest mean pathlength in summer  and the highest values in fall and winter. This might be due to the enhanced storm track activity during fall and winter, where the mass flux (including water vapour) from the troposphere to the stratosphere is largest (Reutter et al., 2015). Further investigations are needed to answer this question.

5     The lower two layers also show a seasonal cycle, however with a smaller amplitude and shorter absolute mean values. While the upper troposphere also shows the shortest mean pathlength in the summer, the minimum in the tropopause is found during winter. Note, due to the large number of very small pathlengths in all seasons and all levels and the logarithmic binning of

*Figure 11: Added sentence regarding the reasons for seasonal changes.*

**Reviewer:** *13. Your final phrase is not a nice end. It directs the reader's attention to ERA5 while the value of your study takes the back seat.*

**Authors:** Thank you for this remark. We adjusted this part:

The influence of the spatial resolution on the results on ISSR pathlengths was shown in this study. The size, magnitude and seasonal cycle of ISSRs are closely related to the underlying spatial resolution of the data. Hence, future studies should focus on three-dimensional data from models for further

10  investigations of physical processes regarding ice supersaturation in the UTLS region.

*Figure 12: Screenshot of reworked final phrase.*

**Reviewer:** *Additional remarks (typos)*

**Authors:** We corrected all typos. Thank you!

[revised manuscript text omitted]

---

## Author Response (AR3)

Dear Farahnaz Khosrawi,

thank you for the acceptance of our manuscript. Please find our technical corrections below. At the end of this document you find a marked-up version of the manuscript.

P4, L2: studies? I guess you rather mean data sets here.

With "studies" we mean the comparison of our results with the studies of Diao et al. (2014) and Spichtinger and Leschner (2016). We clarified this by referring to these studies at the end of the sentence:

*We also compare these results to ERA-Interim as wellas to other studies (Diao et al., 2014; Spichtinger and Leschner, 2016)*

Fig. 2, 3, and 4: These have not been put next to each in one Figure as requested by the referee (at least not in the draft that was submitted). Instead of next to each other, you could also arrange these figures below each other, which may be maybe easier to handle with the printed two column format. Further, the y-axes labelling ("Pressure") should start with a capital letter.

We redesigned the Figs to the following version (exemplary for Fig. 2):

[Figure]

Table 3 and 4: Since you show the same here as for Table 2, but for different parameters, you could write in the Table 3 and 4 "Same as Table 2, but for Water Vapour" and "Same as Table 3, but for RHi [%]", respectively.

We changed that.

Figure 7: typo in in the top panel labelling: stratopshere -> stratosphere

We corrected that.

P14, L7 and L11: space missing between numbers and between % and text, respectively.

We corrected that.

P15, L29: can be also -> can also be

We corrected that.

P16, L21: space between number and unit missing (two occasions)

We corrected that.

P16, L22: degree sign missing

We corrected that.

P17, Fig 8 caption: Add "a", Example of typical flight.... -> Example of a typical flight.....

We corrected that.

P22, Fig 11 caption: Second part of the figure caption does not agree with the figures shown. Which lines?

We corrected that.

P23, L8: distances -> distance

We corrected that.

[revised manuscript text omitted]